# Probabilistic Reasoning with LLMs for Privacy Risk Estimation

**Jonathan Zheng[1]**     **Sauvik Das[2]**     **Alan Ritter[1]**     **Wei Xu[1]**

[1]College of Computing, Georgia Institute of Technology
[2]Human-Computer Interaction Institute, Carnegie Mellon University
jzheng324@gatech.edu

## Abstract

Probabilistic reasoning is a key aspect of both human and artificial intelligence that allows for handling uncertainty and ambiguity in decision-making. In this paper, we introduce a new numerical reasoning task under uncertainty for large language models, focusing on estimating the privacy risk of user-generated documents containing privacy-sensitive information. We propose BRANCH, a new LLM methodology that estimates the $k$-privacy value of a text—the size of the population matching the given information. BRANCH factorizes a joint probability distribution of personal information as random variables. The probability of each factor in a population is estimated separately then combined to compute the final $k$-value using a Bayesian network. Our experiments show that this method successfully estimates the $k$-value 73% of the time, a 13% increase compared to o3-mini with chain-of-thought reasoning. We also find that LLM uncertainty is a good indicator for accuracy, as high variance predictions are 37.47% less accurate on average.

## 1   Introduction

Large language models (LLMs) have shown increasingly strong performance in mathematical and logical reasoning [88, 65, 17], enabling exploration of a broad suite of user-facing applications. One such application is helping users understand the magnitude of privacy risks when disclosing personal information online — a holy grail of usable privacy that remains elusive [63, 40]. For example, how much riskier is it to disclose one's precise age versus a general age category (22 vs. 18-25)? To that end, we introduce a task that instructs LLMs to estimate the $k$ number of people in the world that match the personal attributes or experiences presented in user-written messages on pseudonymous online fora like Reddit [19] or anonymous user interactions with ChatGPT [92, 54]. This task serves both as a new challenge for LLMs' reasoning abilities beyond conventional math and logic tests [30, 49, 36] and as a practical security tool to inform users about online safety.

Traditionally, privacy research has focused on quantifiable properties such as $k$-anonymity [80, 71] applied by *dataset owners* to protect individual records, with privacy risk assessed by the success rate of re-identifying anonymized entries in a database (e.g., a person on Wikipedia) [56, 57, 89]. In this work, we shift the focus to *end-users*, who are neither dataset owners nor curators, by providing an interpretable risk estimate that can be computed *without* access to a comprehensive database containing personal records of all internet users. LLMs must rely on their internal knowledge of demographic statistics from census data to reason about and estimate the identification risk based on the multiple, potentially interconnected, first-person textual disclosures. For instance, LLMs may estimate the privacy risk of a post where a user mentions that she is from Italy, 26 years old, on the spectrum, and has social anxiety (see Figure 1 and 2 for more examples).

Reasoning on these joint attributes remains a significant challenge for language models. In Figure 1, Chain-of-Thought prompting estimates the prevalence rates of social anxiety and autism indepen-

39th Conference on Neural Information Processing Systems (NeurIPS 2025).

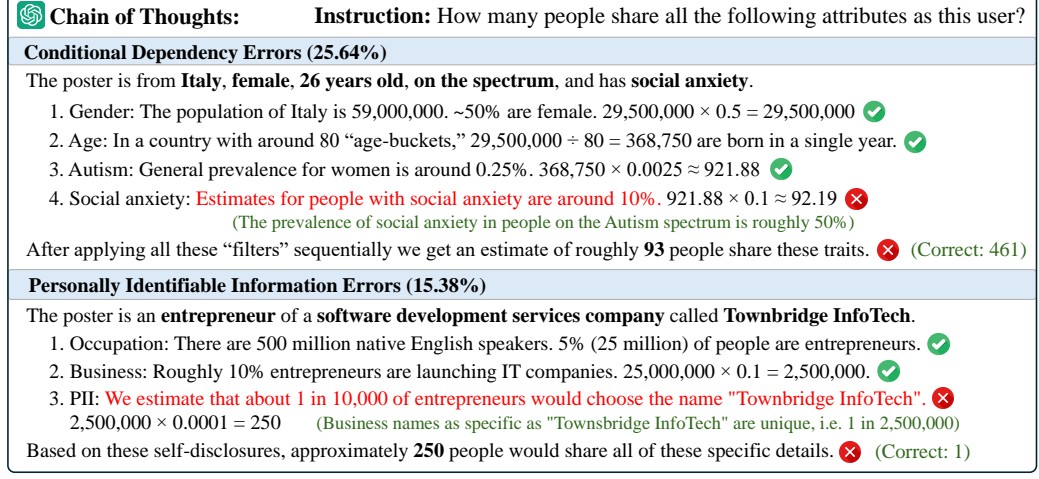

Figure 1: Most common Chain-of-Thought reasoning error types (with occurrence rates) and examples for o3-mini on Privacy Risk Estimation. Errors and correct explanations are highlighted. Chain-of-Thought struggles to model PII and capture relationships between attributes for risk assessments.

dently, failing to account for the conditional dependency whereby being on the spectrum significantly increases the likelihood of social anxiety. To improve privacy risk estimation, we propose BRANCH (see Figure 2), a probabilistic reasoning framework for LLMs that represents each document as a joint distribution of personal attributes. BRANCH first factorizes this distribution by implicitly constructing a Bayesian network, capturing the interdependencies between all of the attributes. Each attribute is then transformed into a textual query and individually estimated using standalone LLMs. Finally, BRANCH reconstructs the joint probability following the structure of the Bayesian network to predict an integer $k$, representing the number of people worldwide who share the relevant personal attributes.

We empirically evaluate BRANCH and state-of-the-art baselines in privacy risk estimation using a new dataset of user posts with human-annotated gold standard values. BRANCH accurately predicts $k$ 72.61% of the time, outperforming baselines by 23.04%. On documents with four or more personal attributes, BRANCH is significantly better than Chain-of-Thought prompting due to its probabilistic model that estimates each attribute individually. BRANCH excels at estimating single-attribute demographics—with low percentage errors from ground truth records—resulting in superior privacy assessments after combining these probabilities. We further evaluate LLM uncertainty as an indicator for estimation accuracy and find that predictions with high variance result in 37.47% lower accuracy.

In summary, privacy risk estimation serves both as a method for evaluating the probabilistic reasoning capabilities of large language models and as an application to support user-centered privacy. Inspired by prior research in the field, we develop a general, human-interpretable value $k$ that helps users understand privacy risks in text, even in the absence of a database of all online users. Our work is also motivated by human-computer interaction (HCI) user studies [40, 19] where participants expressed a desire for explanations on the severity of risks associated with personal disclosures. We envision BRANCH as a practical tool that can provide a number $k$ with a reasoning chain to inform users about the potential identification risks, much like a password strength meter [38], for online privacy.

## 2 Related Work

**Causal and Probabilistic Reasoning in Large Language Models.** Prior research has developed benchmarks to assess LLMs on mathematical and logical reasoning [53, 49, 30, 36], causal reasoning [34, 86], probabilistic reasoning [58, 66, 62], and Bayesian reasoning for general scenarios [21] and human preferences [29]. LLMs have been utilized to construct Bayesian Networks [3] and medical domain Causal Graphs [82]. Statistical inference with Large Language Models has also been applied to decision-making in various domains [93, 21, 46, 44] and constructing in-distribution synthetic samples [11]. Nevertheless, probabilistic reasoning with LLMs remains understudied, particularly for real-world tasks such as modeling privacy risks through Bayesian network elicitation and LLM-based statistical inference. A key component of probabilistic reasoning is uncertainty quantification, which has traditionally been explored in Bayesian neural networks [18] and in deep learning models with dropout [24]. More recently, research has estimated uncertainty in autoregressive models and large

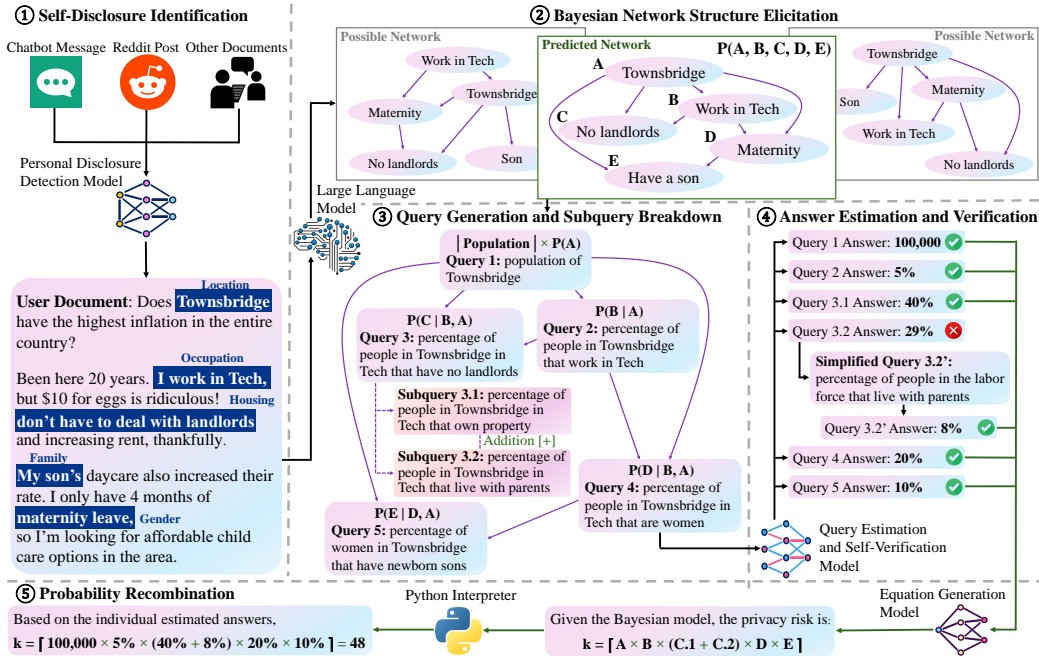

Figure 2: Illustration of the BRANCH framework for estimating the privacy risk $k$ of a document, representing the number of people worldwide who share the personal attributes in the text. LLMs output a single Bayesian Network from the space of possible models for the joint distribution.

language model output with the negative log likelihood of generated sequences [52, 1]. In this work, we use the consistency of multiple LLM generations [84] as an estimate of uncertainty.

There have been substantial approaches to improve LLM capabilities in mathematical and logical reasoning through prompting intermediate reasoning steps [87] and program code [15, 25]. LLM reasoning has been augmented with state evaluators to formulate information as trees [90] or potentially optimizable graphs [5, 94]. Other work has integrated logical constraints [12], code execution [83], and step-by-step supervision [47] during finetuning to enhance general LLM reasoning abilities.

**Privacy Protection and Security.** Canonically, $k$-anonymity has been defined as a property ensuring that each record in a dataset cannot be distinguished from at least $k - 1$ other records based on quasi-identifiers like age and gender [80, 71]. This framework is typically applied by dataset owners who anonymize their data to achieve the desired privacy level. Subsequent enhancements like $l$-diversity [51], $t$-closeness [43], and $\beta$-likeness [13] have addressed limitations of $k$-anonymity to help dataset owners understand and address privacy risks, where complete data visibility enables complex statistical analyses. We flip this problem framing of privacy on its head, focusing not on the dataset owner but the data *contributor*, such as an individual Reddit user who is sharing a post online.

Prior work has utilized LLMs to assess the privacy risks of anonymized database records by attempting to re-identify the original record and measuring privacy success rate [56, 57] and semantic distance to the original record [89]. Quantifying privacy risk for users has been explored before via social media privacy meters [45], but all of these approaches require full access to the dataset of a given platform and do not assess privacy in content disclosures in posts but static profile attributes. Empirical user studies in privacy protection have found that LLMs are useful tools for supporting user privacy [40, 19, 79]. To that end, previous research has used LLMs to identify personal disclosures in user-generated content [2, 42, 91, 39, 81, 7, 27] and rewording the text to protect privacy [19]. However, our approach is novel in that we aim to provide an interpretable quantification of disclosure-based privacy risk for individual data contributors at the point of content sharing.

## 3 Privacy Risk Estimation with Bayesian Network Reasoning

To create contributor-centric privacy risk estimates of user content, we adapt $k$-anonymity —— rather than its more complex derivatives like $l$-diversity [51], $t$-closeness [43], and $\beta$-likeness [13] — because it provides an intuitive and interpretable value ("There are 1,000 other women in Tech in

Townsbridge") that can be meaningfully estimated without access to database records of possible Internet users. To that end, we estimate $k$ relative to an unknown dataset of all real people who might share the characteristics disclosed in a document with BRANCH: **B**ayesian Network **R**easoning for k-**AN**onymity using **C**onditional **H**ierarchies (illustrated in Figure 2). BRANCH is a method that uses LLMs—via prompting or finetuning—to reason probabilistically by constructing a Bayesian Network on a document, which is used to calculate privacy risk with population statistics of personal attributes.

**Problem Setup.** Given a document $U$, we identify the set of disclosures $X$ mentioned in the document and estimate the number of $k$ people that share the same characteristics from the context $C = (U, X)$. Disclosure categories are labeled automatically by a model or manually by humans (more details in §4.1), following the schema introduced in prior work [19]. We model the disclosures in $X$ as random variables $\{X_1, \dots, X_j\}$ and formalize the task to estimate the joint probability $p = P(X_1, \dots, X_j)$ and the population $n$, where the privacy risk estimate is the expected number of people $k = np$ fitting the criteria. In Figure 2, $n$ is the global population while $p$ is the joint probability of a person living in Townsbridge, being a mother, owning a house, and working in Tech.

**Bayesian Network Structure Elicitation.** Directly estimating the joint probability $p$ can be challenging when there are several interconnected personal attributes in a document. To make the privacy risk estimate more tractable and interpretable, BRANCH leverages large language models to implicitly represent the joint probability as Bayesian networks. Each disclosure is modeled as a conditional probability $P(X_i|\{X_{parent}\})$ in the Bayesian network, where LLMs determine parent attributes by first selecting an *ordering* of the disclosures as random variables. While any arbitrary permutation of the disclosures can potentially model the joint probability, BRANCH instructs LLMs to select an ideal ordering based on causality and statistical availability to feasibly estimate $k$. For instance, $P(\text{woman} \mid \text{work in Tech})$ is easier to estimate than $P(\text{work in Tech} \mid \text{woman})$ as gender breakdowns for occupations are available whereas occupation statistics by gender are uncommon.

After determining the variable order, BRANCH determines the *conditional dependencies* of each disclosure. While a fully connected Bayesian network successfully reconstructs the joint distribution via the chain rule of probability, some disclosures in the set $\{X_{parent}\}$ can be independent to simplify model estimation. For instance, in Figure 2, "woman" and "no landlords" are independent when conditioned on "Townsbridge" and "work in Tech" because the gender distribution of an occupation is not impacted by housing status, resulting in the probability $P(\text{maternity leave} \mid \text{work in Tech, Townsbridge})$.

**Query Generation and Subquery Decomposition.** LLMs convert the conditional probabilities into textual queries. An example query for $P(\text{maternity leave} \mid \text{work in Tech, Townsbridge})$ is the *"percentage of Townsbridge residents in Tech **THAT are mothers**"*, where the dependent clause estimates the interest group of "mothers". To lower the variance of $k$, models combine the total population with the first disclosure to generate a specific population query (e.g., $n \cdot P(\text{Townsbridge}) = n' \to \text{"population of Townsbridge"}$). Some probability terms are also decomposed into subqueries for better estimation. For instance, $P(\text{no landlords} \mid \text{Townsbridge})$ contains multiple discrete cases: *"percentage of Townsbridge residents that own property"* **plus** the *"percentage of Townsbridge residents that live with parents"*. BRANCH determines if queries should be decomposed, generates the relevant subqueries, and provides the arithmetic operation in parenthesis to reconstruct the original query.

**Query Estimation and Probability Recombination.** BRANCH instructs LLMs to apply their inherent historical knowledge of demographic statistics to estimate the search queries. Additionally, Retrieval-Augmented Generation can also be used for query estimation (see Appendix M). Some queries may be highly specific due to the conditional dependencies of a disclosure, leading to incorrect probability estimates. To improve the accuracy of the $k$-privacy prediction, LLMs report their confidence of query answers, simplify low-confidence queries (e.g., removing a dependency), and estimate the new queries. Finally, BRANCH uses the constructed Bayesian Network for inference by recomposing the query answers and subquery arithmetic operations to obtain a mathematical equation. A Python interpreter evaluates the final equation to obtain the model privacy estimate $\hat{k}$.

## 4  Experimental Details

### 4.1  Privacy Risk Dataset

**User-LLM conversations and Reddit posts.** We construct a privacy risk estimation dataset of 220 documents: 180 user posts (130 real and 50 synthetic ones) from the pseudonymous online forum

Reddit and 40 real user-LLM conversations from ShareGPT (`sharegpt.com`), annotated with gold labels. This research has been approved by the Institutional Review Board (IRB) with additional measures to safeguard the data, such as replacing all personally identifiable information (PII) with fake identifiers, which are often included in user-LLM conversations. Two in-house annotators first identify and categorize all disclosures (e.g., occupation) in the text based on the schema from prior work [19]. We introduce five new categories, namely emotions, events, crime, possessions, reproductive health, and information about other people, based on insights from a recent user study [40]. The annotators manually construct a Bayesian network from the disclosures, find ground-truth answers to the conditional probabilities, construct a math equation combining the answers, and calculate the final $k$-value for a post. To capture variations in probability orderings and variable dependencies, each post is annotated at least twice, ensuring multiple reasonable interpretations are considered. In total, we annotated 1929 total disclosures for 481 orderings, averaging 139 words and 4.01 disclosures per post, which matches the typical number of natural distribution of personal information across various subreddits [19]. We create a train/test/validation split of 100/66/14 Reddit posts, using all 50 synthetic documents in the train set and evaluating only human written posts, including all 40 user-LLM conversations as test documents to evaluate cross-domain generalization.

**Synthetic Data Generation.**   To share data publicly and securely, we synthetically generate 50 posts[1] that mimic Reddit by prompting LLaMA-3.1-Instruct-8B with few-shot examples, incorporating synthetic PII and severe disclosures related to depression, suicide ideation, mental health, domestic violence, and abortion. To create synthetic data in-distribution with real data, we prompt LLaMA-3.1-Instruct to rephrase user posts and train a T5-large model [69] for style transfer from LLM-generated text back to the style of Reddit. LLaMA-generated data may carry distributional biases, so these posts are also manually inspected to ensure that the synthetic PII is not linked to any real person online and further refined to increase the fidelity of the synthetic data in content and tone using real human reference posts of similar topics. More details are located in Appendix D.

**Threat Model.**   Based on the categories of disclosures in our dataset and the pseudonymous nature of Reddit and ShareGPT, the key threat model [76] we consider is an adversary who may have knowledge of all post-specific disclosure contexts. For instance, a prediction of $k = 1$ for a post with disclosures about location, age, gender, and event-specific circumstances indicates that an adversary with knowledge of that event and who knows a person who matches that location, age, and gender would be able to uniquely identify the poster. This could be someone in their inner circle of friends, for example. It does *not* mean that anyone on the Internet would be able to uniquely identify that poster, as a user's personal attributes are unlikely to be recorded in publicly available databases.

**Ground Truth Validation.**   We assess the quality of our annotation method and dataset by collecting 100 interest groups, averaging 4.32 disclosures per group, from census and public university records that have ground truth populations (e.g., the number of government workers in a city). In-house annotators use our annotation method by utilizing online sources, *excluding* the aforementioned ground-truth databases, to estimate the probability of each disclosure individually and recombine answers to get the population. We report an average percentage error of 22.24% when comparing the reconstructed estimate with the known ground truth population. Notably, the ground truths of the 100 census interest groups exhibit an average error of 24.79% based on the error margins reported in the official census, demonstrating that our dataset consists of high-quality $k$-values comparable to official sources. We also verify annotation quality by doubly annotating 10% of posts. Despite the variability in the Bayesian networks, we find that multiple valid network configurations yield similar estimations of the final k-values. Annotators' predicted $k$-values within the same order of magnitude, yielding high inter-annotator agreement ($\rho = 0.916$), indicating that our method is consistent and robust to differences in network construction.

## 4.2   Evaluation Metrics for Privacy Risk Estimation

We calculate the **Spearman's rank correlation** $\rho$ between LLM outputs $\hat{k}$ and the gold standard $k^*$ and measure the magnitude of error of model predictions. While the ground truth census populations in the validation experiment span up to a couple hundred thousand, our human-annotated $k^*$ in the dataset ranges between 1 and 56,240,520. Given this large variance, we design a **log error metric**:

$$LogError(\hat{k}, k^*) = |\log_2(\hat{k}) - \log_2(k^*)|$$

---

[1]The dataset can be found here.

Privacy is interpreted in log scale as it is analogous to uncertainty in Information Theory [6]. For each user post, there is an equivalence class of $k$ individuals who share the same set of attributes in the text, making any one person's identity indistinguishable from the other $k - 1$. For a random variable $Z$ representing a user's identity, the probability of selecting any single individual is $P(Z = z) = \frac{1}{k}$. The information content of a specific outcome is $I(Z = z) = -\log_2(P(Z = z)) = \log_2(k)$, which measures the uncertainty level for identifying a specific individual in the equivalence class, corresponding to the user's quantified privacy. This log error reflects human assessments of risk, as changes in $k$ at low values result in much larger changes in privacy risk and larger log errors, and variance in model predictions at high $k$-values is not as significant, leading to lower log errors.

We also measure the percentage of model predictions that are outliers compared to the gold standard. We construct intervals to determine if the model output is within an order of magnitude of the gold standard. For $n$ posts, $1 \leq i \leq n$, and hyperparameter $a$, the **range metric** is defined as:

$$RANGE(\{\hat{k}_i\}, \{k_i^*\}) = \frac{1}{n} \sum_{i=1}^{n} \mathbb{1}\left[\frac{\hat{k}_i}{a} \leq k_i^* \leq a \cdot \hat{k}_i\right]$$

This is a general accuracy metric inspired by Brier score [9] that evaluates the percentage of times a model correctly estimates $k^*$ if the predicted $\hat{k}$ falls in a defined range centered around the gold standard. We report $a = 5$ in Table 1 and $a = \{2, 10\}$ in Table 5 in Appendix I.

### 4.3 Baselines and Model Details

For privacy risk estimation, models are instructed to determine $k$ by evaluating the context $C$ and select a subset $X' \subset X$ of personal attributes that can be feasibly estimated. We consider the following baseline approaches:

- **Few-Shot Prompting**: Models are prompted with $j$ demonstrations and asked to estimate the $k$-value of a given post.
- **Chain-of-Thought Prompting (CoT)**: Models are provided with $j$ demonstrations of stepwise reasoning [87] of selecting disclosures, estimating a percentage of people that share each disclosure, and adding this percentage to the final equation to obtain the $k$-value.
- **Program-of-Thought Prompting (PoT)**: Models are provided with $j$ demonstrations of Python functions [15, 25] that create disclosures as variables with estimated percentages and math equations to solve for the $k$-value of a post.

We use GPT-4o [61] and LLaMA-3.1-Instruct 70B for all prompting methods. We run LLaMA-3.1-Instruct-8B model for few-shot prompting and finetuning with LoRA [31], and we use CoT prompting for GPT-4o-mini, o1-preview [59], o3-mini [60], and DeepSeek-r1 [17]. For BRANCH, we prompt LLaMA-3.1-Instruct-70B, GPT-4o, and o3-mini with few-shot demonstrations and finetune LLaMA-3.1-Instruct-8B on the human-annotated train set for 5 components: selecting disclosures, ordering variables, determining conditional dependencies, generating queries and subqueries, and estimating statistics. All models are given $j = 3$ demonstrations, and we use sample decoding with a temperature of 0.7 during inference. To prevent the variability of disclosure detection models from impacting our evaluation, models are provided a user post and the gold standard disclosure and category labels (See Appendix H for details about hyperparameter selections and Q for all the prompts used in BRANCH and the baseline models).

## 5 Experimental Results

### 5.1 Main Results

Table 1 presents the main results and shows that the evaluation metrics correlate strongly with each other. We display the percentage difference in performance on each metric with respect to GPT-4o with chain-of-thought prompting. Our main findings are summarized as follow:

**BRANCH-o3-mini and BRANCH-GPT-4o outperforms all Chain-of-Thought baselines.** On average, these models achieve a Spearman's $\rho$ of 0.856, a log error of 2.08, and is within an order of magnitude of the gold standard 69.79% of the time. Comparatively, the best-performing baseline model is o3-mini CoT, which has a 0.112 lower Spearman's correlation, 0.73 higher log error, and 10.66% lower range metric. We use the paired bootstrap test [4] for $10^6$ iterations to compare

Table 1: Performance table

| Method | Model | All Documents | | | Reddit Documents | | | ShareGPT Documents | | |
|---|---|---|---|---|---|---|---|---|---|---|
| | | $\rho\uparrow$ | Log Err.$\downarrow$ | Range%$\uparrow$ | $\rho\uparrow$ | Log Err.$\downarrow$ | Range%$\uparrow$ | $\rho\uparrow$ | Log Err.$\downarrow$ | Range%$\downarrow$ |
| FEW-SHOT | LLaMA-Instruct-3.1$_{8B}$ | 0.151 | 7.14 | 21.74% | 0.183 | 6.66 | 19.87% | 0.146 | 8.79 | 25.32% |
| | LLaMA-Instruct-3.1$_{8B}$ (FT) | 0.495 | 4.46 | 43.04% | 0.635 | 3.49 | 45.70% | 0.366 | 6.30 | 37.97% |
| | LLaMA-Instruct-3.1$_{70B}$ | 0.435 | 4.86 | 30.00% | 0.255 | 4.85 | 30.46% | 0.590 | 4.89 | 29.11% |
| | GPT-4o (2024-08-06) | 0.565 | 3.94 | 42.17% | 0.519 | 3.55 | 43.05% | 0.608 | 4.69 | 40.51% |
| PROGRAM-OF-THOUGHT | LLaMA-Instruct-3.1$_{70B}$ | 0.615 | 4.03 | 37.83% | 0.519 | 3.82 | 37.09% | 0.656 | 4.44 | 39.24% |
| | GPT-4o (2024-08-06) | 0.673 | 3.33 | 54.78% | 0.624 | 2.99 | 56.29% | 0.678 | 3.97 | 51.90% |
| CHAIN-OF-THOUGHT | LLaMA-Instruct-3.1$_{70B}$ | 0.589 | 3.88 | 46.09% | 0.429 | 3.90 | 43.71% | 0.708 | 3.84 | 50.63% |
| | GPT-4o-mini (2024-07-18) | 0.401 | 5.19 | 37.83% | 0.592 | 3.62 | 43.71% | 0.266 | 8.19 | 26.58% |
| | **GPT-4o (2024-08-06)** | 0.747 | 3.09 | 55.22% | 0.730 | 2.72 | 56.95% | 0.750 | 3.79 | 51.90% |
| | o1-preview (2024-09-12) | 0.761 | 3.06 | 55.66% | 0.727 | 2.82 | 56.95% | 0.779 | 3.52 | 53.16% |
| | DeepSeek R1 (2025-01-20) | 0.724 | 2.98 | 56.96% | 0.685 | 2.64 | 58.28% | 0.737 | 3.62 | 54.43% |
| | o3-mini (2025-01-31) | 0.744 | 2.81 | 59.13% | 0.730 | 2.39 | 64.90% | 0.722 | 3.60 | 48.10% |
| BRANCH (this work) | LLaMA-Instruct-3.1$_{70B}$ | 0.695 | 3.47 | 51.74% | 0.646 | 3.44 | 48.34% | 0.748 | 3.51 | 58.23% |
| | LLaMA-Instruct-3.1$_{8B}$ (FT) | 0.712 | 3.09* | 56.52%‡ | 0.670 | 3.03 | 54.97% | 0.746 | 3.21* | 59.49%‡ |
| | GPT-4o (2024-08-06) | 0.839 | 2.16‡ | 66.96%‡ | 0.797 | 2.16‡ | 66.89%† | 0.871 | 2.18† | 67.09%† |
| | **o3-mini (2025-01-31)** | **0.873** | **1.99*** | **72.61%*** | **0.817** | **2.04†** | **72.19%†** | **0.912** | **1.88*** | **73.42%*** |

Table 1: Performance of different methods on the privacy estimation task. †, ‡, and * represent statistical significance with a p value of 0.05, 0.01, and 0.001, respectively, comparing the BRANCH model to their respective CoT baseline. Cells are colored by percentage $\Delta$ with respect to GPT-4o CoT performance: -15%  -10%  -5%  0%  +5%  +8%  +10%

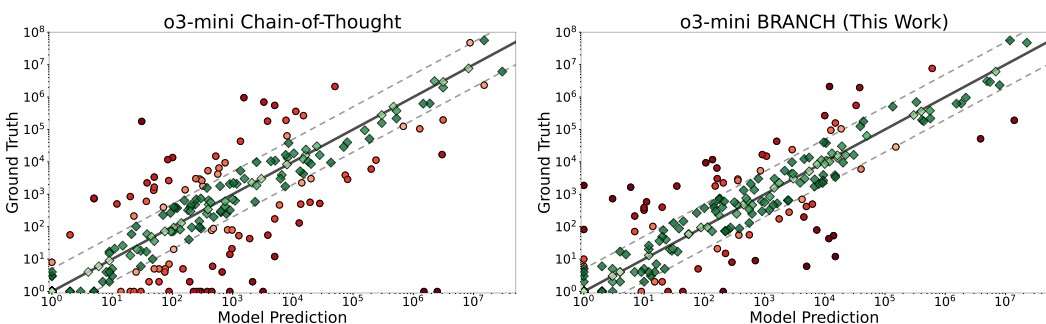

Figure 3: Plots of the model prediction $\hat{k}$ compared to the ground truth $k^*$ in log scale. The dashed lines indicate the acceptable half-magnitude range surrounding the gold standard values. Incorrect predictions are shaded to indicate the level of magnitude of the *residual errors*.

BRANCH to Chain-of-Thought baselines on the same model architecture. Table 1 shows that o3-mini, GPT-4o, and LLaMA-3.1-8B finetuned exhibit a statistically significant improvement in privacy risk estimation when using BRANCH over Chain-of-Thought prompting. In comparison, when prompted with Chain-of-Thought, the mathematical reasoning models R1, o1-preview, and o3-mini are notably worse than BRANCH at reasoning probabilistically over real-world scenarios in this task. We perform an in-depth analysis comparing error modes of Chain-of-Thought and BRANCH in Section 6.

Notably for BRANCH, finetuning LLaMA-3.1-Instruct-8B for each individual module yields 4.78% higher range accuracy than the few-shot prompted baseline of BRANCH LLaMA-3.1-Instruct-70B. GPT-4o with reasoning prompts, R1, and o1-preview are competitive with the finetuned BRANCH model and outperform BRANCH LLaMA-3.1-Instruct-70B by 5% in range accuracy, on average. These models outperform LLaMA-3.1-Instruct due to their better instruction-following capabilities, which is vital for our privacy risk estimation task. LLaMA-3.1-Instruct often ignores instructions, such as providing a percentage answer for a population query.

**BRANCH is safer than Chain-of-Thought prompting.** BRANCH o3-mini and GPT-4o produce lower log errors than their Chain-of-Thought counterparts for 61.74% and 59.57% of posts, respectively. In Figure 3, we plot the model predicted $\hat{k}$ with the gold standard $k^*$ values in log scale for o3-mini with BRANCH and Chain-of-Thought prompting. BRANCH o3-mini predictions are centered along the main line $\hat{k} = k^*$, whereas o3-mini Chain-of-Thought contains more outliers, including conservative predictions that provide little utility to users as privacy estimation tools. o3-mini and GPT-4o baselines classify 28.48% of $k$-values as high-estimate outliers—serious errors where the models significantly underestimate the privacy risk of a document. These mispredictions can mislead users about their online safety by portraying high-risk situations as low-risk. For example, in Figure 3, o3-mini Chain-of-Thought significantly underestimates privacy risk to be $\hat{k} = 2,500,000$ despite

the true value being $k^* = 1$. In comparison, BRANCH GPT-4o and o3-mini only exhibit 12.17% of cases where they underestimate privacy risk, with those errors being much smaller in magnitude.

**BRANCH is generalizable to different domains and can account for highly sensitive personal information.** Table 1 presents the separate results of the Chain-of-Thought baselines and the BRANCH models on each domain, Reddit and ShareGPT. Notably, Chain-of-Thought prompting baselines struggle significantly more on User-LLM conversations than on Reddit posts, with o3-mini CoT producing 16.80% lower range accuracy. This is especially concerning given that User-LLM conversations frequently contain personally identifiable information (PII), making robust and accurate modeling crucial to capture serious privacy risks. In contrast, BRANCH not only outperforms the chain-of-thought baselines in both domains, our method improves when evaluating posts with severe privacy risks, yielding better results on User-LLM conversations compared to user-written posts with 3.96% higher range accuracy and 0.085 higher Spearman's correlation, on average.

### 5.2 BRANCH Component Analysis

**LLM predictions of demographic statistics.** We investigate the strength of BRANCH for privacy risk estimation by individually evaluating the capabilities of LLMs for estimating individual statistics and modeling Bayesian networks in Figure 4. Using human-written single attribute probability queries (e.g., "percentage of people in Townsbridge that are in Tech") with ground truths, we calculate the average percentage error of model probability estimates to the ground truth. Both o3-mini and GPT-4o demonstrate strong capability in estimating individual attributes, with average percentage errors of 16.60% and 17.45%, respectively—lower than the average margin of error found in census data. BRANCH performance is similar across both domains, demonstrating that the models' capability in generating accurate demographic statistics translates to strong performance in privacy risk estimation.

**Bayesian structure modeling.** Prior work on Bayesian structure modeling uses score-based methods [72] to evaluate all polynomial orderings and conditional dependencies. Instead, we evaluate the structure of Bayesian networks in BRANCH with human-annotated Bayesian networks in Figure 4 and find that BRANCH has an average Kendall's $\tau$ rank correlation of 0.59 and 0.39 for Reddit and ShareGPT, respectively, indicating moderate agreement. For modeling conditional dependencies, we find that disclosures are, on average, conditionally independent of 42.80% of prior variables in BRANCH and 45.81% in human-annotated networks, suggesting that BRANCH approximates human reasoning in Bayesian network elicitation for modeling privacy risk.

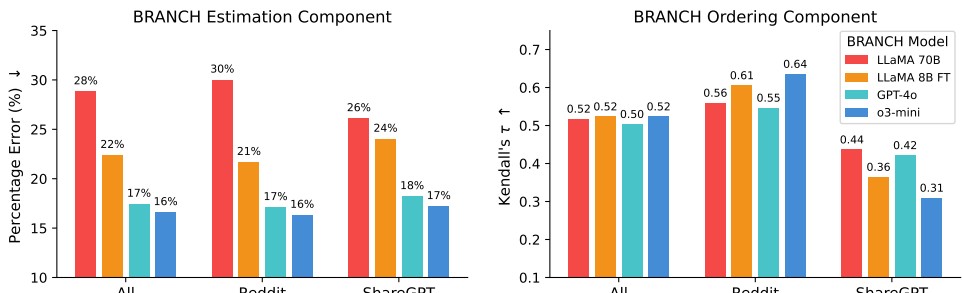

Figure 4: Analysis of the **individual** components in BRANCH for query estimation and Bayesian model ordering. Query estimation is evaluated with percentage error against ground truth queries and answers, and model ordering is evaluated with Kendall's $\tau$ against human-made Bayesian networks.

## 6  Comparing Chain-of-Thought and BRANCH

To better understand how BRANCH and Chain-of-Thought prompting scale with input complexity, we compare GPT-4o and o3-mini performance across posts with varying numbers of personal disclosures. For posts with fewer than 4 disclosures, the range accuracy of both methods is roughly similar, with BRANCH achieving 67.16% and Chain-of-Thought achieving 61.28% on average for both models. For posts with 4 or more disclosures, the performance gap between BRANCH and Chain-of-Thought prompting widens substantially. In BRANCH, o3-mini and GPT-4o increase their range accuracies to 74.22% and 69.53%, respectively, while Chain-of-Thought prompting decreases o3-mini and GPT-4o range accuracy to 55.47% and 52.34%, respectively. For both models, BRANCH has higher accuracy

on complex posts than it does on the entire dataset of user posts, indicating that the Bayesian network modeling step in BRANCH enables models to handle more personal attributes in a document.

**Chain-of-Thought errors occur due to the lack of probabilistic modeling.** Figure 1 presents example posts where BRANCH correctly and Chain-of-Thought incorrectly estimates privacy risk. We analyze GPT-4o and o3-mini CoT errors and find that 25.64% of errors are due to the lack of dependencies modeled between disclosures. For instance, o3-mini uses 10% for the "prevalence of social anxiety", not accounting for being "on the spectrum" which increases social anxiety prevalence to 50%. 20.51% of CoT errors occur due to the lack of subqueries that precisely estimate probability, and 15.38% of errors occur due to PII, like the unique business name "Townsbridge InfoTech" which results in $k = 1$. While BRANCH accurately captures this information in the Bayesian network, Chain-of-Thought underestimates the privacy risk $k = 250$ by assuming 1 in 10,000 people use this business name. In contrast, BRANCH errors arise from issues also common in Chain-of-Thought prompting, with 38.47% of CoT baseline errors and all BRANCH errors attributable to incorrect probability estimations or attribute selections. Thus, BRANCH outperforms Chain-of-Thought prompting on complex posts specifically due to its probabilistic methodology that can model conditional dependencies and personally identifiable attributes.

## 7 Sampling and Uncertainty

In Section 5, we use sampling as the decoding method to generate point estimates for $k$ values. To assess the variability of reasoning processes for the privacy risk estimate, we **re-evaluate** models $n$ times for privacy risk estimation to construct mean point estimates. Specifically, we run GPT-4o with BRANCH and Chain-of-Thought prompting 5 times for privacy risk estimation to obtain mean $\bar{k}$ estimates for all user posts from the test set in Reddit. We also use **self-consistency** to re-sample only the query estimation component in BRANCH and combine the expected answer $\bar{p}_i$ for all queries in a post to construct $\bar{k}$. We compare $\bar{k}$ with the gold standard $k^*$ using the evaluation metrics defined in Section 4.2. We report the variability of predictions using the coefficient of variation, calculated as the standard deviation of the $k$ values normalized by their mean. For self-consistency, we define the total variance as the sum of the individual variances across all queries within a post.

| Model Ranges | Coeff. Variation | Spearman's $\rho\uparrow$ | Log Err.$\downarrow$ | Range %$\uparrow$ | Range$_{\text{Low Var.}}$ %$\uparrow$ | Range$_{\text{High Var.}}$% $\uparrow$ |
|---|---|---|---|---|---|---|
| CoT Re-Evaluation | 1.072 | 0.691 | 3.17 | 49.67% | 61.63% | 33.85% |
| BRANCH Self-Consistency | 0.854 | 0.812 | 2.08 | 68.21% | 71.28% | **63.16%** |
| BRANCH Re-Evaluation | 1.013 | **0.838** | **1.91** | 68.21% | **86.08%** | 48.61% |

Table 2: GPT-4o Mean Estimate Results for **Re-Evaluating** the model and using **Self-Consistency** on the query estimation module in BRANCH. The average **Coeff**icient of **Variation** (CV) is the normalized standard deviation of the predicted $k$ values. Range accuracy is categorized as high or low variance depending on whether the predicted $k$'s CV is above or below the mean.

**BRANCH mean estimates are more accurate than individual $k$ estimates**. As seen in Table 2, BRANCH GPT-4o performance for the $\bar{k}$ estimates increases, resulting in 68.21% range accuracy. Re-running the entire BRANCH pipeline yields the best performance, achieving 0.25 lower log error than using one run of BRANCH GPT-4O. However, re-sampling the entire BRANCH model results in significantly higher variability, with a coefficient of variation of 1.013, compared to 0.854 achieved by applying Self-Consistency only to the query estimation component. The low variance of the query estimation module in BRANCH under self-consistency, together with its comparable performance to a single BRANCH run, suggests that the pipeline is robust to fluctuations in LLM-estimated demographic statistics, as the LLMs used in BRANCH are internally consistent when estimating these quantities within our dataset domains. In comparison, re-evaluating Chain-of-Thought prompting is much less consistent for privacy risk estimation, as the mean $\bar{k}$ estimate yields 5.55% lower range accuracy and 0.44 higher log error than one run of CoT prompting. Chain-of-Thought also has the highest coefficient of variation of 1.072, indicating that it is not only less reliable for accurate privacy risk estimation but also significantly more inconsistent across runs.

**The implicitly constructed Bayesian Networks in BRANCH are stable across repeated executions of the pipeline**. The overall performance of the BRANCH model remained stable across 5 runs of the entire pipeline, suggesting robustness in the Bayesian network elicitation process itself. To further quantify structural consistency, we compute the mean Structural Hamming distance (SHD) between the induced shared subgraphs across different BRANCH runs, which measures the number of edge

operations–additions, deletions, or reversals–needed to transform one Bayesian network to another. The mean SHD across all user posts was 1.29, with a minimum of 0 and a maximum of 10. By comparison, the mean maximum possible SHD (i.e., comparing a fully connected subgraph to an entirely disconnected one) is 11.36, further affirming that the constructed Bayesian network structures in BRANCH are highly consistent across runs despite potentially noisy or missing priors. These findings collectively suggest that the BRANCH pipeline produces stable and consistent Bayesian network structures, even under variability in LLM outputs.

**LLMs are well-calibrated for privacy risk estimation.** Following insights from [84], we use model variance as a measurement of estimation uncertainty. To evaluate this, we stratify the range accuracy based on whether the coefficient of variation for each post is above or below the mean. We find that predictions of $k$ with high variance are significantly more likely to be incorrect, with re-evaluating the entire BRANCH model resulting in 86.08% accuracy on low variance estimates, compared to 48.61% for high variance privacy estimates. Chain-of-Thought prompting exhibits a similar trend, achieving 27.78% higher accuracy on posts with high certainty. This discrepancy is less pronounced when using self-consistency on the query estimation module, as BRANCH shows comparatively strong performance even on high-variance examples with 63.16% accuracy. Nevertheless, model uncertainty is a useful indicator for identifying when privacy risk estimations are likely to be inaccurate.

## 8 Conclusion

We present a new probabilistic reasoning task for Large Language Models: quantifying the privacy risk of documents containing personal attributes. To make risk assessments interpretable, we develop a $k$ value for privacy risk, representing the number of individuals who share the specific combination of attributes mentioned in the text. We develop BRANCH, a new methodology that uses Bayesian Networks to model a joint distribution over personal attributes. By decomposing the joint probability into a set of conditional probabilities, BRANCH enables more accurate and tractable estimation of the privacy risk. We construct a dataset annotated with gold standard values and find that BRANCH can accurately estimate the $k$ value 72.61% of the time. In our analyses, we demonstrate that BRANCH outperforms baseline models on complex inputs with several interconnected attributes due to its probabilistic decomposition and ability to accurately estimate statistics of individual attributes. Finally, we show that the variance in predicted $k$-values serves as a proxy for model uncertainty—where higher variance reliably signals less accurate $k$ estimates. Our results show that BRANCH can be a practical tool to inform users about potential privacy risks when sharing information online.

## Acknowledgements

We would like to thank Joseph Thomas for his help in constructing the $k$-anonymity dataset, refining synthetically generated posts, and annotating for the privacy risk value. This work is in part supported by the NSF CAREER Awards IIS-2144493 and IIS-2052498. Any opinions, findings, and conclusions or recommendations expressed in this material are those of the authors and do not necessarily reflect the views of the National Science Foundation. We would like to thank Microsoft's Azure Accelerate Foundation Models Research Program for providing computational resources to support this work.

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

# A  Limitations

We construct a comprehensive dataset capturing various topics and domains of user posts with sensitive personal information. Due to the cost of annotating privacy risk in text, we intentionally focused on the natural distribution of atomic user posts on domains rich in personal attributes that map directly to structured and publicly available ground truth sources (e.g., census data, university records), namely real-world user-written Reddit posts and user-LLM conversations. That said, we acknowledge that these domains may not capture the full diversity of real-world scenarios. Future work could look at developing user studies to evaluate BRANCH in more real-world contexts and exploring additional threat models and contexts (e.g., an individual who also knows additional information not present in the post) to better understand BRANCH's robustness across use cases. Cross-domain generalization is also an important direction, including applications to Amazon reviews, job forums, and other online platforms where personal information is frequently shared. These efforts will help investigate how well BRANCH can generalize to diverse, real-world scenarios.

The distribution of disclosures in our dataset is usually between 3–6 per post and rarely extends past 8 disclosures, which closely matches the natural distribution of personal information in these domains. While we find that BRANCH outperforms CoT-based methods in handling dense attribute interactions as the number of disclosures increases, we have not extensively tested posts with dozens of disclosures. Future work could estimate the privacy risk of text that contains more than 10 personal disclosures, such as with the post history of a Reddit user.

In our current setup, we are assessing how closely LLM-based methods approximate human experts. Human gold standard labels are validated based on doubly annotating a portion of our dataset and with the validation experiment that shows that for interest groups similar to those found in Reddit and ShareGPT, the human annotation methodology exhibits very low error when compared to the ground truths. However, we acknowledge that finding a perfect ground truth is unattainable for some cases, which is an inherent limitation of working with real data in this setting. Nevertheless, the authors of this paper manually review the annotation process to ensure the quality of the gold k value.

Our method BRANCH utilizes large language models to estimate privacy risk. This can pose a computational burden, particularly because our methodology requires multiple LLM inference steps to construct a joint probability, create queries, and estimate individual answers. The computational burden may also increase when utilizing model uncertainty as an indicator of accuracy. To reduce this burden, future work could look at:

1. LLM Output Caching: By caching intermediate outputs, especially for frequently seen or structurally similar inputs, we can significantly reduce the computation time.
2. Lightweight Model Substitution: In many cases, such as simpler posts, Chain-of-Thought outputs or lightweight, distilled models could be used to approximate certain stages of the BRANCH pipeline without full LLM calls.
3. Pipeline Optimization: Batching and combining BRANCH steps to reduce the total number of LLM calls required to estimate privacy risk.

In principle, with a substantially larger set of human-annotated data (albeit requiring significant upfront human annotation effort), it would also be possible to fine-tune an end-to-end model that eliminates some intermediate components and significantly reduces inference-time computational cost.

To ground our methodology and establish an upper bound on potential privacy risk, we focus on a strong adversary with access to knowledge of all post-specific disclosure contexts. This setup enables a rigorous evaluation of the maximum possible privacy leakage under realistic but constrained assumptions. Future work could study user-personalized threat modeling, where privacy risk is estimated relative to the knowledge scope of specific, personalized adversaries. In these cases, users can specify the types of personal information that their threat model has access to (e.g., the knowledge that a coworker has about a user). We also plan on looking at general threat models with differing levels of personal access, such as quantifying privacy risk to a stranger who does not have access to personal information outside of publicly available databases.

The primary purpose of our model is to provide users with a supplementary tool to quantify the privacy risk associated with privacy-sensitive documents and online personal disclosures. While the failure cases of the best-performing models tend towards overprotection as they estimate low $k$-values for a post, there are some instances where models will misclassify a document to have a lower privacy risk than in reality. Prior user studies have been conducted in LLM usage for privacy protection, but we have not conducted a specific user study to test our model for providing quantifiable privacy risk estimates. We plan to deploy user studies in the future to explore the utility of our model, the best ways to present this information to users, and how users adjust behavior in response.

While we do not observe obvious biases against specific demographic groups in our dataset, we have not studied this extensively, as this requires significantly more data and human annotation resources than what is feasible within the scope and budget of this academic study. For evaluation, we compare the LLM-generated estimations to human-annotated ground truth in Section 5.2, ensuring that LLM-internal biases do not affect the model evaluation. Future research could conduct a comprehensive audit of all possible sources of bias (e.g., gender, race, geography, education level, age, culture) and investigate whether there could be potential bias from the LLM-internal demographic priors utilized in our method, which could impact its use as a tool for certain demographics. This analysis should also determine if failure cases unevenly affect certain groups, especially marginalized or vulnerable populations, before these $k$-estimation models can be deployed to real users.

# B   Impact Statement

This research and the data sharing plan were approved by the Institutional Review Board (IRB) at our institution. In our study, we sample from Reddit and ShareGPT following established data curation practices and treat LLM responses as approximate statistical surrogates. We take several measures to safeguard the personal information in our dataset. In accordance with IRB policy, we anonymized all of the user posts and user-LLM conversations collected during the study by replacing all instances of personally identifiable information (PII), such as names and emails, with synthetic data. For these real documents and synthetically generated posts, we perform a rigorous manual inspection to ensure that the synthetically generated PII in our dataset does not correspond to any real individual. For instance, we verify that no online search results can be found for a synthetically generated name. In our dataset, we explicitly collect a portion of user posts with personal disclosures from marginalized or vulnerable populations. We inspect our dataset to verify that the texts, titles, topics, and personal disclosures included in these posts contain no potential harm, such as hate speech or biases that may be perpetuated, that would impact these populations. All examples shown in the paper are either synthetically created by humans or by large language models and accurately reflect the real data. We hired in-house student annotators ($18 per hour) over crowd workers for annotation. Every annotator was informed that their annotations were used to create a privacy risk estimation dataset. For our synthetically generated posts, we take several precautions to ensure the fake user posts are natural while not posing any real harm. To prevent misuse by the aforementioned threat models, we will not release our dataset of real user posts and LLM conversations and finetuned models to the public. We will instead only share them upon request with researchers who agree to adhere to the following ethical guidelines:

1. Use of the dataset is limited to research purposes only.
2. Redistribution of the models and dataset without the authors' permission is prohibited.
3. Compliance with Reddit's official policies is mandatory.

Additionally, posts that have been deleted by users will not be provided at the time of the data request. To request access to these resources, please email the authors.

The primary purpose of our model is to provide users with a supplementary tool to quantify the privacy risk associated with privacy-sensitive documents and online personal disclosures. In cases where the models incorrectly predict the privacy risk, these model failures do not pose an additional risk to user privacy. While the failure cases of the best-performing models tend towards overprotection as they estimate low $k$-values for a post, there are some instances where models will misclassify a document to have a lower privacy risk than in reality.

Overall, we believe there are many potential positive social benefits of our work. By developing tools to quantify the privacy risk of users for sensitive documents, we can inform users of potential cybersecurity risks and enable these users to make more informed decisions when sharing personal information online. As mentioned in the introduction, we envision this tool to serve as something akin to a password strength meter [38] for indicating the privacy risks of online disclosures. While no user studies have been conducted on this $k$-anonymity task, we plan to investigate the utility of this tool for informing users about the risks of content sharing of personal information. Specifically, we envision this privacy risk quantification being utilized in a "Grammarly for Privacy"-like browser extension that helps users identify privacy-sensitive disclosures and quantify the overall risk of these disclosures, and that suggests user-acceptable rephrasings [19, 40] that preserve the semantic meaning of the original disclosure but are less privacy-sensitive — i.e. have higher predicted $k$-values.

## C  Further Related Work

**Demographic Estimation.**  There has been considerable work in political science and statistics on estimating the size of groups of people in a population sharing certain characteristics. Prior work uses multilevel regression and post-stratification methods to accurately model populations with overlapping characteristics like political opinion and race in national surveys and political polls [41, 10, 64, 16]. Other research utilizes probability and statistical theory to interpret the proportions as probabilities $p$ for demographic subgroups [37, 22, 73, 67] and develop point estimates $np$ for the number of people in subgroups using $n$ as the population size.

**Privacy Protection in NLP.**  Research in NLP has found that LLMs engender several privacy risks by, for example, generating personally identifiable information and email addresses unintentionally or explicitly if prompted [50, 78, 32, 14, 55, 74]. LLMs are also capable of inferring private user attributes from text, including age, sex, and location [77]. Prior work has pre-trained LLMs with differential private data to develop privacy-aware models [68, 75, 70], and privacy enforcing algorithms have been developed in text or structured data to uphold the privacy properties of $k$-anonymity [26] or $\beta$-likeness [28], resisting machine learning-based attacks.

## D  Synthetic Dataset

| Metric | Train | Test | Dev |
|---|---|---|---|
| Number of Total Posts | 100 | 106 | 14 |
| Number of Orderings | 216 | 230 | 35 |
| Number of disclosures | 905 | 889 | 135 |
| Number of Domains | 30 | 22 | 5 |
| Average Length of Posts | 795.41 | 727.48 | 905.36 |
| Average Disclosures / Ordering | 4.19 | 3.87 | 3.86 |
| % of Posts in Shared Subreddits | 49% | 22% | 0% |

Table 3: Details about the Train, Test, and Development Splits of the privacy risk dataset.

The synthetic posts and the annotation process for obtaining the $k$ value of the posts can be found here. To construct the synthetic posts, we utilize the Reddit API to obtain natural posts on the subreddits r/oakland, r/TwoXChromosomes, r/mentalhealth, r/cheyenne, r/gatech, and r/chapmanuniversity. Additionally, we used keyword searches to obtain mental health-related and women's health-related posts for more serious topics, such as "abortion", "birth control", "domestic violence", "depression", "suicide", and "bipolar disorder". 1000 posts were obtained from each subreddit, and the posts were manually filtered for posts that had at least 4 disclosures or other inferrable personal attributes.

We construct logically consistent user profiles by randomly sampling for personal attributes, such as age, gender, occupation, educational major, and personal experience. We use the natural posts as samples for user attribute distributions. For instance, an attribute distribution instance of age, gender, personal experiences, and mental health is created for a natural post containing only mentions of those disclosures. User profile distributions are randomly sampled, and the personal attributes are randomly modified. LLaMA-3.1-Instruct-8B is provided with a sampled user distribution and 3 natural posts as few-shot examples and is instructed to create a post that is consistent with the provided profile. The posts are immediately inspected for personally identifiable information (PII) and manually removed if existing PII in the synthetic post is related to a real person, community, or website.

We finetune T5-large for informal style transfer for 10 epochs, 500 warmup steps, and a weight decay of 0.01. Due to the large size of Reddit posts, we use a batch size of 1. Training takes at most 3 hours. For inference, we use a higher temperature of 0.9, as we find that this temperature results in typos and inconsistencies in the text similar to those found naturally on Reddit. This temperature also resulted in a higher degree of informality that was natural on Reddit. In general, the percentage of words that were misspelled after using style transfer was 6%, which matched the spelling error rates on the subreddits that the synthetic posts were based on. The subreddits r/mental_health and r/TwoXChromosomes were very formal, having a spelling error rate of 2% compared to other Reddit communities with a misspelling rate of 21%, as these subreddits were often forums where people were seeking serious advice or discussion. We evaluate the fidelity of these posts by providing a sample of 100 synthetic posts and 100 real posts to in-house annotators. For synthetic posts, the in-house annotators have a classification accuracy of 52%, and 50 mislabeled posts are included in our dataset for $k$-estimation.

To improve the fidelity of the synthetic posts further, in-house annotators were given synthetic posts and explicitly instructed to modify the posts to seem more natural. Annotating a single post took 20 to 30 minutes. Reddit posts obtained from the respective subreddits were also provided as reference posts for tone, content, and writing. The in-house annotators were instructed to write down changes and comments that indicated if a post seemed unnatural. Only 2% of our synthetic Reddit posts require drastic tuning of the text, as most of the edits made were made regarding the style of the posts. In general, the common errors of LLaMA-3.1-Instruct-8B are:

- **Too formulaic**: The post endings were composed in a very formulaic manner. In general, the posts were structured as a personal story, before ending with 2-3 questions and an expression of gratitude. Posts were manually edited to remove questions and develop different endings based on the references.

- **Repetition**: Posts often exhibited an unnatural amount of repetition. For instance, if a post mentioned the phrase "I just feel like", or "I'm tired of" at least once, LLaMA-3.1-Instruct-8B would be inclined to repeat these phrases dozens of times in a post. Posts were made to be more concise and natural.

- **Common Expressions**: LLaMA-3.1-Instruct-8B often produced very common expressions from post to post. Examples include "Has anyone else experienced this?" and "I feel like I'm not really living" for depression issues are very common. Posts were edited to restructure these expressions.

- **Incongruity**: Posts were sometimes generated with logical errors, such as time inconsistencies, where a post would mention "today" but describe something that happened last week. Other posts were generated with conflicting personal attributes, such as mentioning being a first-year and a third-year in college. Some posts had sentences with tones inconsistent with the remainder of the text. Posts were reviewed to remove these logical and tonal inconsistencies.

- **Vagueness**: Some sentences were vague, resulting in unnatural text. Examples include the phrases "I feel like I'm just stuck in this rut" and "I'm thrown into this hamster wheel going nowhere." These phrases are removed or modified to add additional details relevant to the user.

- **Redundancy**: There are some sentences that would repeat the main idea of a post, which was notably unnatural according to in-house annotators. For instance, for a post about someone being in a domestic violence situation, "I feel like I'm trapped in this cycle of abuse and I don't know how to escape" is redundant as their personal relationships were already mentioned. These sentences were removed.

- **First-Person Perspective**: LLaMA-3.1-Instruct-8B often had issues with perspective in generating questions. For instance, unnatural questions like "Domestic violence survivors, how did you get out of it?" are generated in a post about someone suffering from domestic violence. This is much more unnatural than asking "How do I get out of this?" Questions are edited to be clearer and more grounded in the perspective of the user and the topic of the user post.

To evaluate the effectiveness of the synthetic posts, we also evaluate a finetuned version of BRANCH without synthetic posts in the training set. We observe a drop in range accuracy to below 40%. In contrast, models of the same size finetuned with the synthetic data show substantial performance gains over both prompt-based inference and training on real data alone, suggesting that this synthetic augmentation strategy supports generalization rather than hinders it.

# E   K-Estimation Dataset

Table 3 shows the details of the train, test, and dev split of our $k$-estimation dataset. Table 4, Table 5, and Table 6 provide the domain breakdown for the training set, development set, and test set, respectively. On average, annotating a single user post for two or more orderings takes an hour to complete, and it took 30 minutes to review and revise the annotations. As each post is annotated multiple times, the orderings result in different $k$ values, with differing high levels of certainty. In-house annotators are all college-educated and fluent in English.

We follow prior work [19] and consider the following personal disclosure categories for privacy risk estimation: (1) location, (2) age, (3) relationship status, (4) gender, (5) pet, (6) appearance, (7) race/nationality, (8) sexual orientation, (9) health, (10) family, (11) occupation, (12) mental health, (13) emotions, (14) reproductive health, (15) finance, (16) education, (17) crime, (18) events, (19) disclosure of other people, and (20) personally identifiable information.

Given a user post, annotators are instructed to identify and categorize all disclosures based on the augmented annotation schema. Annotators must generate a number $k$ that matches the number of people sharing personal disclosures, including the subreddit name (e.g. r/twoxchromosomes for gender or r/cheyenne for location). Annotators model the task as a probability/proportion estimation task and utilize the probability chain rule to decompose joint probabilities into conditional probabilities. The annotators manually construct a Bayesian network from the disclosures by determining a probability ordering for the self disclosures for broadest to most specific and by causality. To construct the Bayesian network structure, annotators determine the conditional dependencies of each disclosure with respect to the prior disclosures in the ordering. The conditional probability terms are then converted into textual queries and valid subqueries, and annotators find ground-truth answers to these queries based on publicly available statistics and census data. For subqueries, annotators additionally annotate for the existence of parentheses and the specific mathematical operation to combine the subqueries together. Finally, annotators construct a math equation combining the answers and calculate the final $k$-value for a post. To capture variations in probability orderings and variable dependencies, each post is annotated at least twice, ensuring multiple reasonable interpretations are considered.

| Domain | # of Posts |
|---|---|
| r/Salem | 6 |
| r/chicago | 3 |
| r/oxforduni | 2 |
| r/relationship_advice | 2 |
| r/antiwork | 1 |
| Total | 14 |

Table 5: Domain breakdown for the **dev** set.

| Domain | # of Posts |
|---|---|
| r/mentalhealth | 12 |
| r/TwoXChromosomes | 12 |
| r/gatech | 11 |
| r/chapmanuniversity | 10 |
| r/Cheyenne | 8 |
| r/oakland | 6 |
| r/manchester | 4 |
| r/roswell | 3 |
| r/eauclaire | 3 |
| r/oaklanduniversity | 3 |
| r/durham | 2 |
| r/jacksonville | 2 |
| r/kansascity | 2 |
| r/NYCApartments | 2 |
| r/duke | 2 |
| r/smithcollege | 2 |
| r/udub | 2 |
| r/uchicago | 2 |
| r/atlanta | 1 |
| r/southampton | 1 |
| r/bostoncollege | 1 |
| r/emersoncollege | 1 |
| r/gvsu | 1 |
| r/asktransgender | 1 |
| r/breastfeeding | 1 |
| r/EntrepreneurRideAlong | 1 |
| r/MeetNewPeopleHere | 1 |
| r/motorcycle | 1 |
| r/PhR4Friends | 1 |
| r/relationship_advice | 1 |
| Total | 100 |

Table 4: Domain breakdown for the train set.

| Domain | # of Posts |
|---|---|
| *Reddit* | 66 |
| r/Cheyenne | 5 |
| r/gatech | 5 |
| r/chicago | 5 |
| r/ucla | 5 |
| r/Salem | 5 |
| r/BostonU | 5 |
| r/hoodriver | 5 |
| r/mit | 5 |
| r/twoxchromosomes | 5 |
| r/mentalhealth | 5 |
| r/oakland | 3 |
| r/oxforduni | 2 |
| r/venting | 2 |
| r/MeetPeople | 2 |
| r/antiwork | 1 |
| r/CPTSD | 1 |
| r/dating_advice | 1 |
| r/Mounjaro_ForType2 | 1 |
| r/personalfinance | 1 |
| r/SpicyAutism | 1 |
| r/SuicideWatch | 1 |
| *ShareGPT* | 40 |
| **Total** | **106** |

Table 6: Domain breakdown for the **test** set.

Annotators are further instructed to annotate for the reliability of the ground truth source. For instance, reliable ground truth sources come from educational institutions or census databases, while informal surveys are labeled as unreliable. For each query, annotators also utilize ChatGPT [61] to obtain a prediction value for each query. Annotators record ChatGPT's response and the reliability and annotator confidence of the prediction compared to the ground truth value. If ChatGPT does not return a source, an alternate Google source is used to create a prediction. Finally, disclosure queries are labeled based on the availability of ground truths. If there is either a reliable or an unreliable ground truth for a disclosure and associated query, the query is labeled as feasible. If there is no ground truth, but the disclosure is vital to highlight or estimate (e.g., having two knee injuries) for privacy risk, the query is labeled as important. If there is no ground truth and the disclosure is relatively unimportant (e.g., feeling sad), the query is labeled as infeasible.

We conduct validation experiments to evaluate the human annotations of $k$ and find that the average percentage error of populations of interest is 22.24%. Notably, this percentage error is calculated over interest population groups of size $\geq 4$, which reflects the typical populations covered in our dataset and in the domains we cover. While compounded errors of probability estimates are a theoretical concern, their practical impact remains low under our annotation protocol, especially for the domains (e.g., Reddit, ShareGPT) that utilize census information or public databases.

# F BRANCH Details

The disclosures are ordered based on broadest to most specific and causality (i.e., disclosures that are valid subgroups of other disclosures are ordered last). For instance, an age disclosure is ordered before a health disclosure, as age is a determining factor for several health conditions. Additionally, the disclosures are ordered based on how much the conditional distribution changes from the joint probability. For instance, the probability $P(\text{male}, \text{hip injury} \mid 24, \text{Atlanta})$ is wildly different from the joint distribution $P(24, \text{male}, \text{hip injury} \mid \text{Townsbridge})$, as it is unlikely for a 24-year-old to have a hip injury compared to older ages. Comparatively, $P(24, \text{hip injury} \mid \text{Townsbridge}, \text{male})$ is much more similar to the joint distribution as the gender distribution of hip injuries is much more uniform, so gender occurs before age in the probability ordering.

For BRANCH o3-mini, we make slight changes to the pipeline. For the disclosure selection prompt, we find that o3-mini will often select all of the personal disclosures due to its increased reasoning capabilities. We modify the prompt to emphasize the selection of disclosures that are feasibly estimated by humans with limited access to online databases. For the query generation module, o3-mini tends to generate hyperspecific queries that result in many overestimations of privacy risk with $k = 1$. To mitigate the repetition of information, we provide the model with the history of queries generated for a post and instruct the model to generate a new non-redundant query that does not repeat information generated in the existing queries. To further reduce the impact of hyperspecific queries, we utilize a confidence thresholding method for query simplification. After the answer estimation module, o3-mini provides a confidence score for its answer. If the confidence score is lower than a specified value, the generated query is too specific and difficult to estimate, so we instruct o3-mini to simplify and estimate the new query. We set the threshold as 0.55 and run the threshold-simplification loop once for each query.

LLaMA-3.1-Instruct performs worse on BRANCH due to its instruction following capabilities. As such, we also modify the BRANCH pipeline accordingly:

- The disclosure selection module output is checked to ensure that the included disclosures are only for disclosures provided in the prompt to prevent hallucinations.
- The query generation module is repeatedly run until a query is generated with the words "percent", "population", or "number".
- The answer estimation module output is checked to ensure that the answers are at least 1 for population queries and less than 1 for percentage queries.
- Models are not prompted to generate subqueries for discrete cases involving addition, as LLaMA-3.1-Instruct-70B often generates subqueries that sum up to be greater than 1.

For non-reasoning models, we use a temperature of 0.7 for sample decoding, as we find that running BRANCH with a temperature of 0 decreases range metric performance by 5.41%, on average. For finetuning BRANCH, we train Llama-3.1-Instruct-8B with LoRA on 4 A40 GPUs for 10 epochs, with a total batch size of 16, which takes at most 6 hours per run. We use a learning rate of 3e-4 and AdamW [48] as the optimizer with a weight decay of 0.01. We use a cosine learning rate schedule with a warmup ratio of 0.03, and for LoRA hyperparameters, we set rank=8, alpha=32, and dropout=0.05. We update the embedding layer during finetuning as well. Overall, we finetune 5 modules of BRANCH: selecting disclosures, ordering disclosures, determining conditional dependencies, query generation, and query estimation. The query generation method is finetuned to optionally produce subqueries, and we do not train a query simplification module for this finetuned variant. The query answers are recomposed with a Python interpreter.

# G Metrics

We experiment with using percentage error to evaluate model predictions compared to gold standards and find that percentage error is much more susceptible to the influence of outliers compared to log error. For instance, the percentage error of $k^* = 2$ and $\hat{k} = 100$ is 4900%. We additionally find that the percentage error is inaccurate and does not agree with the other metrics for evaluating systems, as it evaluates GPT-4o Chain-of-Thought to have an average percentage error of 113157%. By comparison, the LLaMA-3.1-Instruct-70B Chain-of-Thought yields an average percentage error of 26208%, roughly 5 times better than GPT-4o, even though the model is worse in every other metric. Additionally, percentage error is not as interpretable for $k$, especially for low $k^*$ values. For $k^* = 2$ and $\hat{k} = 100$, the log error is 5.64, indicating that the model is 5.64 bits off from the gold standard $k^*$. Privacy is evaluated by humans on a logarithmic scale, so log error, which represents the average residuals between the model predicted $\hat{k}$ and the gold standard $k^*$, is more interpretable for humans. For the legend in Table 1 and Table 7, percentage error is computed by the difference between the inverse of the log errors (e.g., $\frac{1}{5.64}$) to fit the differences in proportion for Spearman's $\rho$ and Range accuracy. For selecting the hyperparameter $a$, we found that $a = 5$ matched human assessments of privacy risk the best, following feedback from our in-house annotator. For instance, for a privacy risk $k = 500$, a prediction between $[100, 2500]$ was found to be the most reasonable. We also report $a = 2$ and $a = 10$ in Figure 5.

## H   Experimental Details

We selected GPT-4o and o3-mini as models for BRANCH due to their strong baseline performances. Hyperparameters were tuned using the development set of our dataset: a confidence threshold of 0.55 provided the highest range accuracy and prevented the model from over-simplifying or under-simplifying queries; temperature values of 0 and 1 were tested but underperformed compared to 0.7. Finally, we used 5 reruns for uncertainty estimation as a practical compromise, balancing robustness with the computational cost of BRANCH's multi-step pipeline.

We also test linear regression models for predicting $k$. We obtain LLaMA-3.1-8B [20] dense vector embeddings, which are then optionally embedded with the sparse auto-encoder `llama-3-8b-it-res` [35]. We use the dense or sparse vector to predict $k$ by fitting a linear regression model.

For o1-preview and DeepSeek R1, we experiment with regular few-shot prompting, zero-shot prompting, and program-of-thought prompting on 30 samples. We find that all of these variants result in lower performance across all three metrics compared to chain-of-thought prompting, with more than 10% lower accuracy. For the finetuned baseline models, LLaMA-3.1-Instruct-8B is finetuned with the same parameters as BRANCH. Specifically, we use LoRA with 4 A40 GPUs for 5 epochs and a total batch size of 16. Finetuning takes at most 4 hours. We use a learning rate of 3e-4 and AdamW [48] as the optimizer with a weight decay of 0.01. We use a cosine learning rate schedule with a warmup ratio of 0.03, and the LoRA hyperparameters are the same as in finetuning BRANCH. The embedding layer is also updated. For finetuning the RoBERTa and LLaMA-3.1-Instruct-8B baseline models, we train models to classify the base 10 bin values ($10^{\lceil \log_{10}(k) \rceil}$) as the $k$-number for a given user post and disclosures. We also test regression models instead of classifying with base-10 bins and find that in practice, LLaMA-3.1-Instruct-8B and RoBERTa-large perform much more poorly in the regression setting than in the classification setting. We attribute this performance decrease to the limited amount of data that we have for training. Due to the large range of possible $k$-values, performance is much more difficult when requiring the model to predict millions of potential $k$-values compared to the closest base-10 bin.

We test LLaMA-3.3-70B Instruct for all 3 prompting baselines and for a prompting variant of BRANCH. We also test an End-to-End reasoning model of BRANCH, similar to the reasoning chains of o1-preview, R1, and o3-mini. Using the human process for annotating models, we finetune LLaMA-3.1-8B-Instruct with LoRA to produce the individual chain-of-thought steps of selecting and ordering disclosures, determining the conditional independence assumptions, generating a textual query, estimating the answers to the queries, and recombining the answers to produce a final $k$ value all in one inference step. This model setup mimics the thought process of humans, as well as the reasoning thread and final output produced by reasoning models.

| Model | a = 2 | a = 5 | a = 10 |
|---|---|---|---|
| LLaMA-3.1-8B Dense Linear | 13.04% | 19.13% | 20.87% |
| LLaMA-3.1-8B Sparse Linear | 11.30% | 14.35% | 17.83% |
| RoBERTa-large Finetuned | 13.48% | 30.00% | 40.43% |
| LLaMA-3.1-Instruct$_{8B}$ Few Shot | 12.61% | 21.74% | 30.00% |
| LLaMA-3.1-Instruct$_{8B}$ (FT) | 23.48% | 43.04% | 51.74% |
| LLaMA-3.1-Instruct$_{70B}$ Few Shot | 12.17% | 30.00% | 40.00% |
| LLaMA-3.1-Instruct$_{70B}$ PoT | 20.87% | 37.83% | 51.74% |
| LLaMA-3.1-Instruct$_{70B}$ CoT | 26.52% | 46.09% | 56.96% |
| LLaMA-3.3-Instruct$_{70B}$ Few Shot | 12.61% | 29.13% | 36.52% |
| LaMA-3.3-Instruct$_{70B}$ PoT | 23.04% | 42.17% | 54.35% |
| LLaMA-3.3-Instruct$_{70B}$ CoT | 22.61% | 45.22% | 56.57% |
| GPT-4o-mini (08-06) CoT | 22.61% | 37.83% | 49.57% |
| GPT-4o (08-06) Few Shot | 18.70% | 42.17% | 54.78% |
| GPT-4o (08-06) PoT | 31.74% | 54.78% | 66.09% |
| GPT-4o (08-06) CoT | 32.17% | 55.22% | 62.61% |
| o1-preview CoT | 33.48% | 55.66% | 63.91% |
| DeepSeek R1 CoT | 32.17% | 56.96% | 68.70% |
| o3-mini CoT | 33.91% | 59.13% | 66.96% |
| BRANCH Finetuned End-to-End | 27.39% | 49.13% | 62.61% |
| BRANCH LLaMA-3.1$_{70B}$ Few Shot | 29.57% | 51.74% | 58.70% |
| BRANCH LLaMA-3.3$_{70B}$ Few Shot | 31.30% | 53.48% | 62.61% |
| BRANCH LLaMA-3.1$_{8B}$-Finetuned | 32.17% | 56.52% | 67.39% |
| BRANCH GPT-4o (08-06) Few Shot | 41.74% | 66.96% | 78.26% |
| BRANCH o3-mini (01-31) Few Shot | 40.43% | 72.61% | 80.87% |
| Human Agreement (10% of posts) | 51.43% | 78.79% | 85.71% |

Figure 5: Comparison of different $a$ hyperparameter values for the range metric across all the models tested on our dataset.

## I   Further Results

We evaluate the doubly annotated user posts with the $k$ metrics as a theoretical human upper bound of performance. We use one of the author's annotations as the gold standard and treat the other in-house annotator as the predictions. In our doubly annotated data, we find that the variance mainly arises from the differing Bayesian networks that are constructed to solve for k. Human agreement achieves a Spearman's $\rho$ of 0.916, log error of 1.57, and range metric of 78.79%, indicating that annotators generally agree within the same order of magnitude and that $k$ prediction models are still significantly short of human annotators.

Table 7 provides the results for the additional model setups on our $k$ estimation task. The linear classifiers perform poorly with an average Spearman's $\rho$ of -0.018, log error of 10.17, and range metric of 8.94%, indicating

| Model | All Documents | | | Reddit Documents | | | ShareGPT Documents | | |
|---|---|---|---|---|---|---|---|---|---|
| | $\rho\uparrow$ | Log Err.↓ | Range%↑ | $\rho\uparrow$ | Log Err.↓ | Range%↑ | $\rho\uparrow$ | Log Err.↓ | Range%↓ |
| LLaMA-3.1-8B Dense Linear | -0.031 | 10.56 | 19.13% | -0.062 | 11.26 | 9.27% | -0.075 | 9.23 | 37.97% |
| LLaMA-3.1-8B Sparse Linear | -0.149 | 9.07 | 14.35% | 0.025 | 9.07 | 8.61% | -0.341 | 11.71 | 25.32% |
| RoBERTa-large Finetuned | 0.339 | 5.17 | 30.00% | 0.547 | 3.90 | 37.75% | 0.056 | 7.59 | 15.19% |
| LLaMA-3.3-Instruct$_{70B}$ Few Shot | 0.385 | 4.88 | 29.13% | 0.227 | 4.66 | 34.44% | 0.546 | 5.31 | 18.99% |
| LLaMA-3.3-Instruct$_{70B}$ PoT | 0.578 | 4.00 | 42.17% | 0.411 | 3.97 | 41.72% | 0.679 | 4.07 | 43.04% |
| LLaMA-3.3-Instruct$_{70B}$ CoT | 0.595 | 3.80 | 45.22% | 0.374 | 4.12 | 44.37% | 0.792 | 3.20 | 46.84% |
| BRANCH LLaMA-3.1-Instruct$_{8B}$-(FT) End-to-End | 0.625 | 3.58 | 49.13% | 0.562 | 3.49 | 47.68% | 0.640 | 3.75 | 51.90% |
| BRANCH LLaMA-3.3-Instruct-70B Few Shot | 0.754 | 3.17$^\dagger$ | 53.48%$^\dagger$ | 0.668 | 3.34$^\dagger$ | 49.67% | 0.827 | 2.85 | 60.76%$^\dagger$ |

Table 7: Performance of additional methods on the $k$ estimation task. † indicates a statistical significance value of $p < 0.05$ when comparing BRANCH to Chain-of-Thought using the same base model. Cells are colored by percentage $\Delta$ with respect to GPT-4o CoT performance:

| -15% | -10% | -5% | 0% | | +5% | +8% | +10% |
|---|---|---|---|---|---|---|---|

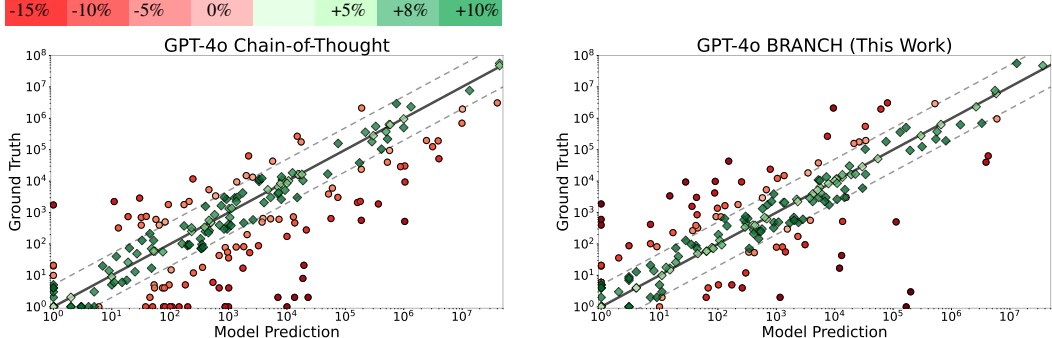

Figure 6: Plots of the GPT-4o prediction $\hat{k}$ compared to the ground truth $k^*$ in log scale. The dashed lines indicate the acceptable half-magnitude range surrounding the gold standard values. Incorrect predictions are shaded to indicate the level of magnitude of the *residual errors*.

that a linear relationship between disclosures and privacy risk does not exist. The finetuned RoBERTa-large is on par with LLaMA-3.1-Instruct-70B Few-Shot prompting with a range accuracy of 30.00%. For LLaMA-3.3-70B Instruct, we find that model performance is on par with LLaMA-3.1-Instruct-70B for all the prompting variants, and the BRANCH model performs slightly worse than GPT-4o Chain-of-Thought. The comparable performance of LLaMA-3.3 with LLaMA-3.1 indicates that our $k$-estimation task is difficult across models of similar architectures, and new training data from recent text does not yield better probabilistic reasoning capabilities. BRANCH LLaMA-3.3-Instruct results are also statistically significant to LLaMA-3.3-Instruct with Chain-of-Thought prompting. We also evaluate the disclosure detection capabilities of models by not providing GPT-4o with the gold standard labeled disclosures. GPT-4o Chain-of-Thought without the disclosure list drops to 3.27 log error and 49.67% range accuracy, indicating the importance of disclosure spans for $k$-estimation.

The BRANCH end-to-end finetuned model only yields a Spearman's $\rho$ of 0.562, a log error of 3.49, and a range metric of 47.68%, producing a lower performance compared to BRANCH LLaMA-3.1-Instruct-70B with few-shot prompting. We attribute this poor performance to the complexity of the task. Given the number of intermediate steps in $k$ estimation, LLaMA-3.1-Instruct-8B is incapable of producing the BRANCH process end-to-end, resulting in lower performance compared to BRANCH pipeline variants. Additionally, due to having only 216 training points, there is limited training data to learn the process end-to-end.

Figure 6 provides the dot plots for comparing the Chain-of-Thought prompting with BRANCH for GPT-4o. As seen with o3-mini, BRANCH GPT-4o is more centered around the main diagonal, predicting more values in the acceptable range. However, BRANCH GPT-4o also has some underestimation outliers similar to GPT-4o CoT.

Figure 7 provides a detailed breakdown of the errors of the models for $k$ estimation. The log error magnitude and percentage of occurrence for all of the overestimation and underestimation errors is reported separately for each model. Importantly, BRANCH GPT-4o and o3-mini have some of the lowest rates of underestimation errors, with only o1-preview being better. However, o1-preview overestimates $k$ much more, resulting in more limited utility in the model. Our BRANCH models are the best at not underestimating privacy risk while maintaining utility by accurately predicting $k$. Figure 5 presents the range metric with the hyperparameters $a = 2$ and $a = 10$. For $a = 2$, we find that the BRANCH o3-mini accuracy drops to 40% and increases to 81% for $a = 10$. The relative performance between models remains consistent regardless of the hyperparameter $a$.

## J Ablation Studies

We further analyze the impact of certain modules utilized in BRANCH in Table 8. We utilize all of the unaffected intermediate output of the original BRANCH module. We analyze the impact of the conditional dependencies

| Model | All Documents | | | Reddit Documents | | | ShareGPT Documents | | |
|---|---|---|---|---|---|---|---|---|---|
| | $\rho\uparrow$ | Log Err.$\downarrow$ | Range%$\uparrow$ | $\rho\uparrow$ | Log Err.$\downarrow$ | Range%$\uparrow$ | $\rho\uparrow$ | Log Err.$\downarrow$ | Range%$\downarrow$ |
| GPT-4o Fully Disjoint Bayes Network | 0.755 | 2.96 | 54.78% | 0.661 | 3.33 | 48.34% | 0.848 | 2.26 | 67.09% |
| GPT-4o Fully Connected Bayes Network | 0.797 | 2.40 | 62.61% | 0.776 | 2.23 | 65.56% | 0.817 | 2.72 | 56.96% |
| GPT-4o No Subqueries Module | 0.837 | 2.30 | 63.91% | 0.772 | 2.44 | 60.93% | 0.860 | 2.02 | 69.62% |
| o3-mini No Simplification Module | 0.850 | 2.16 | 68.70% | 0.813 | 2.19 | 68.21% | 0.873 | 2.11 | 69.62% |

Table 8: Model performances of differing BRANCH GPT-4o ablation studies. We utilize all of the intermediate output in the original BRANCH model for better comparison. Cells are colored by percentage $\Delta$ with respect to the original BRANCH performance: -15% -10% -5% 0% +5% +8% +10%

selection process in constructing an accurate Bayesian network for BRANCH-GPT-4O. We construct fully disjoint Bayesian networks for each user document, which assumes all personal disclosures to be independent of each other, and fully connected Bayesian networks, where disclosures are dependent on all of the prior disclosures. We instruct LLMs to generate new queries, estimate answers, and recombine the probabilities. BRANCH GPT-4o with fully independent disclosures results in a range accuracy of 54.78%, and a log error of 2.96, and BRANCH GPT-4o with fully dependent disclosures results in a range accuracy of 62.61%, and a log error of 2.40. Interestingly, we find that the fully disjoint Bayesian network performs poorly on Reddit and well on ShareGPT, achieving range accuracies of 48.34% and 67.09%, respectively. On the other hand, the fully connected Bayesian network performs well on Reddit and poorly on ShareGPT, achieving range accuracies of 65.56% and 56.96%, respectively. Thus, we find that the conditional dependence classification module in BRANCH is critical to high performance, as BRANCH outperforms both ablations by producing queries that accurately represent the conditional probability terms while being easier to estimate for both the domains Reddit and ShareGPT.

We analyze the impact of the optional modules of BRANCH GPT-4o. 12.5% of queries are broken down into subqueries, which primarily consist of age range queries, which estimate the percentage of people that are in a certain range and are divided by the size of the range. We use BRANCH without the subquery breakdown and find that performance drops, resulting in 0.837 Spearman's $\rho$, 2.30 log error, and 63.91% range metric, indicating that the subquery does result in easier query estimation. We note that BRANCH still outperforms all of the baseline models, even without this optional node. For the simplification module, we find that only 6.56% of GPT-4o answers are classified as incorrect and are simplified. In comparison, 35.86% of o3-mini answers are simplified due to low certainty. Removing the simplification module, we find that performance also drops slightly with a Spearman's $\rho$ of 0.851, log error of 2.16, and range

| Model | % overestimate | % underestimate |
|---|---|---|
| LLaMA-3.1$_{8B}$ Dense Linear | 44.78% (11.35) | 36.09% (14.81) |
| LLaMA-3.1-8B Sparse Linear | 52.61% (10.71) | 33.04% (12.88) |
| RoBERTa-large (FT) | 34.35% (6.44) | 35.65% (7.34) |
| LLaMA-3.1-Instruct$_{8B}$ Few Shot | 25.22% (7.98) | 53.04% (9.76) |
| LLaMA-3.1-Instruct$_{8B}$ (FT) | 12.17% (5.67) | 44.78% (7.44) |
| LLaMA-3.1-Instruct$_{70B}$ Few Shot | 49.57% (7.16) | 20.43% (4.77) |
| LLaMA-3.1-Instruct$_{70B}$ PoT | 42.17% (6.10) | 20.00% (5.45) |
| LLaMA-3.1-Instruct$_{70B}$ CoT | 34.78% (6.82) | 19.13% (5.84) |
| LLaMA-3.3-Instruct$_{70B}$ Few Shot | 46.09% (7.51) | 24.78% (4.38) |
| LLaMA-3.3-Instruct$_{70B}$ PoT | 34.35% (6.89) | 23.48% (5.32) |
| LLaMA-3.3-Instruct$_{70B}$ CoT | 31.74% (6.33) | 23.04% (5.70) |
| GPT-4o-mini (08-06) CoT | 10.87% (4.94) | 51.30% (8.32) |
| GPT-4o (08-06) Few Shot | 27.39% (5.79) | 30.43% (6.21) |
| GPT-4o (08-06) PoT | 14.78% (4.84) | 30.43% (6.89) |
| GPT-4o (08-06) CoT | 12.17% (4.20) | 32.61% (6.36) |
| o1-preview CoT | 36.09% (5.76) | 8.26% (6.11) |
| DeepSeek R1 CoT | 26.96% (4.96) | 16.09% (6.87) |
| o3-mini CoT | 16.52% (4.99) | 24.35% (5.86) |
| BRANCH (FT) End-to-End | 29.57% (6.62) | 21.30% (5.41) |
| BRANCH LLaMA-3.1$_{70B}$ Few Shot | 25.22% (6.31) | 23.04% (6.13) |
| BRANCH LLaMA-3.3$_{70B}$ Few Shot | 25.65% (6.14) | 20.87% (5.52) |
| BRANCH LLaMA-3.1$_{8B}$ (FT) | 16.52% (6.64) | 26.96% (5.47) |
| BRANCH GPT-4o (08-06) Few Shot | 20.87% (4.56) | 12.17% (5.62) |
| BRANCH o3-mini (01-31) Few Shot | 15.22% (4.70) | 12.17% (4.88) |

Figure 7: Detailed error breakdown of models on our dataset. The log error of overestimations and underestimations of privacy risks are shown in parentheses.

accuracy of 68.70%, indicating that utilizing verbalized certainty scores in o3-mini can improve performance by simplifying specific queries.

## K  BRANCH Error Analysis

We perform an error analysis on BRANCH o3-mini and GPT-4o. For each incorrect privacy risk prediction, we manually identify and categorize the error reason. We find that all errors occur due to one of three reasons: (1) attribute selection, (2) query generation, or (3) probability estimation. Notably, these errors also occur frequently in Chain-of-Thought prompting errors, which occur 38.47% of the time. For BRANCH, 20.63% of o3-mini errors and 25.33% of GPT-4o errors occur due to disclosure selection, where BRANCH selects too few disclosures for estimation. 46.03% of errors in o3-mini and 37.33% of errors in GPT-4o emerge from the query generation component, as BRANCH may often generate too specific queries that are difficult to estimate accurately. These errors often further stem from overly strong conditional independence assumptions. 33.33% of

errors in o3-mini and 37.33% of errors in GPT-4o occur due to the probability estimation component, as models may underestimate a simple yet specific query due to the lack of pre-training data, like the percentage of people who are social media managers to a financial coach.

For underestimation errors, we also perform an analysis for the severity of this error. We define the error to be **minor** severity if the actual privacy risk $k$ is greater than 50, **moderate** severity if the actual privacy risk is between 10 and 50, and **major** severity if the privacy risk is less than 10. We find that the majority of underestimation errors in o3-mini are minor, with 53.85% for o3-mini and 50.00% for GPT-4o being on posts where the gold standard privacy risk is above 50. 15.38% and 14.29% of the errors of o3-mini and GPT-4o respectively are moderate severity, and the remaining 30.77% and 28.57% of o3-mini and GPT-4o underestimation errors are major. Given that BRANCH only underestimates error 12.17% of the time, our model has a significant error rate of less than 4%, indicating the strength

| Model | % Correct |
|---|---|
| GPT-4o CoT | 52.34% |
| o1-preview CoT | 57.81% |
| DeepSeek-R1 CoT | 57.81% |
| o3-mini CoT | 55.47% |
| BRANCH GPT-4o CoT | 69.53% |
| BRANCH o3-mini CoT | **74.22%** |

Figure 8: Percentage of model predictions within the acceptable range of $k^*$ for posts with 4 or more personal disclosures.

of our method. Moreover, we find that 83.33% of major severity underestimation errors are predictions with high variance, further allowing us to filter out these errors as uncertain model predictions.

## L  Comparing Chain-of-Thought to BRANCH

Figure 8 shows the performance of Chain-of-Thought prompting baselines and BRANCH GPT-4o and o3-mini on difficult posts with 4 or more personal disclosures. We mainly utilize the range accuracy metric to determine when the model is correct or not and compare the individual predictions between both methodologies. The best performing baseline model is o1-preview and R1 with 57.81% accuracy, while BRANCH o3-mini is the best overall model, correctly predicting 74.22% of posts with 4 or more personal disclosures, which is 1.61% higher than its comprehensive range accuracy. BRANCH GPT-4o also outperforms the baseline models, successfully predicting the privacy risk for 69.53% of complex posts.

We further conduct a comparison between BRANCH and Chain-of-Thought Prompting for o3-mini and GPT-4o, where the performance discrepancy is the largest. For posts where BRANCH correctly predicts $k$ to be within the acceptable range but Chain-of-Thought prompting does not, the average number of disclosures in these posts is 4.44 and 4.26 for o3-mini and GPT-4o, respectively. 66.10% of these posts in BRANCH o3-mini and 64.91% of these posts in BRANCH GPT-4o have 4 or more personal disclosures. By comparison, for user posts where the Chain-of-Thought models correctly predict $k$ to be within the acceptable range but BRANCH does not, the average number of disclosures in these posts is 3.68 and 3.6 for o3-mini and GPT-4o, respectively. Only 53.57% of these posts predicted by o3-mini and 50% of these posts predicted by GPT-4o have 4 or more disclosures.

## M  Retrieval Augmented Generation

We also evaluate retrieval-augmented generation (RAG) systems by modeling query estimation as an attributed question-answering task [8] with evidence. To compare with BRANCH, we utilize the pre-existing output (e.g., determining the conditional dependencies) of the **GPT-4o** pipeline. We create Retrieve-Then-Read systems by using the Google Search API as a document retriever for each individual query. GPT-4o is instructed to select relevant information from the documents to estimate the answer or utilize its standalone knowledge to estimate the query if no information is found. We provide GPT-4o with 10, 20, 50, and 100 snippets of the relevant documents of a query, the entire HTML text of the top-10 documents, and 10 re-ranked snippets using BM25 on the 100 retrieved documents. We also use the commercial retrieval-augmented AI search engine Perplexity-AI, namely `sonar`, `sonar-pro`, `sonar-reasoning`, `sonar-reasoning-pro`, and `sonar-deep-research`. The results of the RAG systems are reported in Figure 9.

**Retrieval Augmented Generation systems do not sufficiently outperform BRANCH**. The best system, `sonar-deep-research`, only marginally improves performance for GPT-4o, achieving 0.839, log error of 2.16, and a range accuracy of 67.39%. 64% of RAG systems perform worse than BRANCH-GPT-4o across all metrics, which contrasts with the strong performance of AI agents that use information retrieval to achieve state-of-the-art results on benchmarks like GAIA [85, 33, 23]. We attribute these results to the niche queries created in BRANCH, as 71.30% of queries have no document evidence in the first 10 search-retrieved documents. To remedy this, we simplify all the generated queries and run RAG with Google and `Perplexity-sonar`, which still yields lower results than BRANCH with 3.48% lower range accuracy and 0.25 higher log error. RAG systems perform worse on User-LLM conversations than on User Posts, with 14.29% and 35.71% of systems exhibiting a log error of 0.32 or higher on Reddit and ShareGPT documents, respectively.

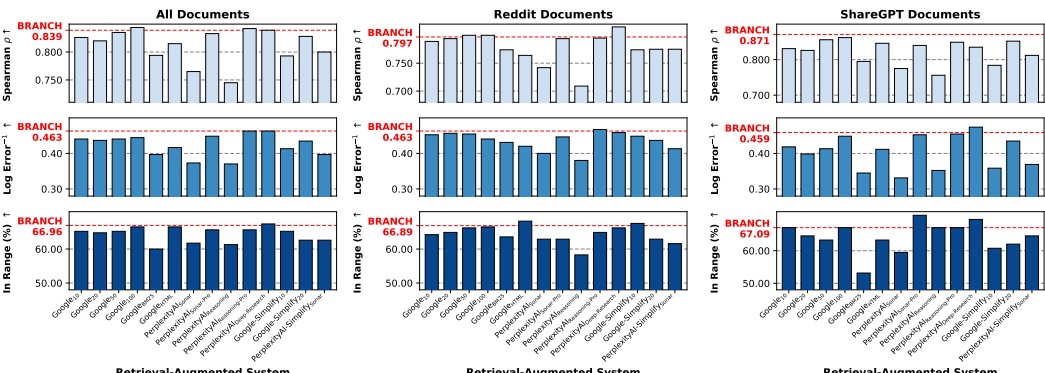

Figure 9: Privacy risk estimation with retrieval-augmented generation systems. BRANCH GPT-4o is highlighted in red.

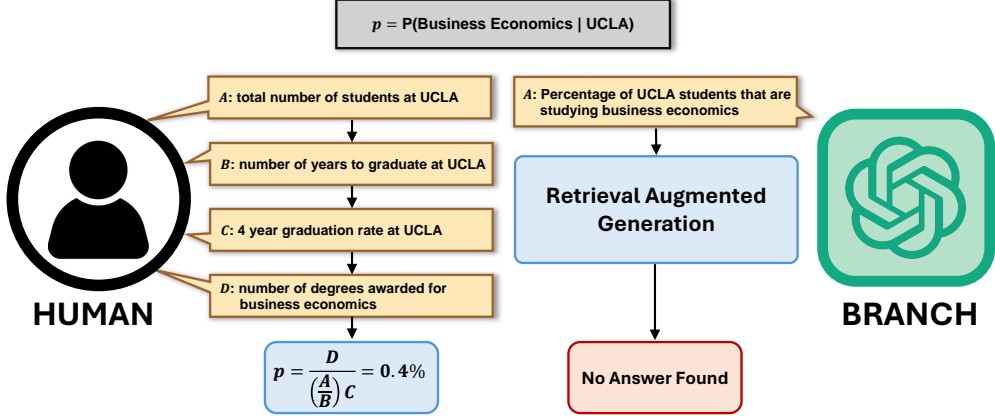

Figure 10: Example of a query and the information retrieval process between RAG systems and humans. Retrieval Augmented Generation systems often fail to find evidence for a generated query in BRANCH. Information retrieval is also difficult for humans, as ground truths for probabilities are often found through multiple steps that break down disclosures into multiple search queries.

Overall, `sonar-deep-research` is the best RAG system, improving range accuracy and matching Spearman's $\rho$ and log error compared to BRANCH GPT-4o. The two models exhibit high agreement with a Spearman's $\rho$ of 0.942, and the percentage of overlapping data points where both models are in the range of $k^*$ is 84.42%. `sonar-deep-research` predicts a larger $\hat{k}$ than BRANCH for 65.22% of posts and produces a more even error distribution than BRANCH GPT-4o by underestimating and overestimating privacy risk 15.65% and 16.96% of the time, respectively.

On average, 67.11% of user posts with incorrect BRANCH predictions are also incorrect with RAG because BRANCH often produces niche queries. We prompt GPT-4o to determine if there is evidence for a query in the first 10 search-retrieved documents and find that 71.30% of queries result in no document evidence. Figure 10 shows an example failure case of Retrieval Augmented Generation. This information retrieval step is hard even for humans, requiring multiple information retrieval steps to adequately calculate a ground truth for a probability. Documents also negatively affect RAG performance; for example, "the percentage of college students that are second-years" (25%) is wrongly influenced by one snippet describing a "90% first-year retention rate" at a particular university. In another example, one snippet had the statistic that 1.3% of women experienced domestic violence in the past year (based on a 2000 report), which resulted in the percentage of women in the United States who have experienced domestic violence to be estimated as 1.3%. In reality, this metric is much higher, ranging between 25% and 30%. Retrieval Augmented Generation also results in incorrect query formats, as it will mix up percentage and population queries because the provided snippets may only provide available percentages for a population query and only provide numbers for a percentage query.

To improve retrieval accuracy, we create an information retrieval baseline that assumes all disclosures to be independent, resulting in simple queries with answers that can be searched with retrieval. This baseline with 10 retrieved documents is the worst-performing systems, resulting in 0.652 Spearman's $\rho$, 3.02 log error, and 54.78% range accuracy, demonstrating that simple retrieval systems cannot predict the $k$ value of users, as the conditional dependencies of disclosures that result in niche queries are vital for $k$-estimation performance.

| Model Ranges | Precision↑ | Recall↑ | Macro-F1↑ | Interval Size |
|---|---|---|---|---|
| CoT Re-Sampling | 25.78 | **83.02** | 24.11 | 12.70 |
| BRANCH Self-Consistency | 33.49 | 45.53 | 25.60 | **3.80** |
| BRANCH Re-Sampling | **34.49** | 74.89 | **32.03** | 11.00 |

Table 9: GPT-4o Mean Estimate and Prediction Interval Results for **Re-Sampling** the model and using **Self-Consistency** on the query estimation module in BRANCH. The interval size is in $\log_2$.

# N   Prediction Intervals of Privacy Risk Estimation

To help provide a more informative range of possible privacy risk values to users, we utilize LLM re-sampling of Chain-of-Thought and BRANCH GPT-4o to construct $k$ prediction intervals using $\bar{k}_i$ and standard deviation $\sigma_i$, defined as $\left[\max\{1, \bar{k}_i - 2\sigma_i\}, \bar{k}_i + 2\sigma_i\right]$ since $k$ cannot be lower than 1. We also use **self-consistency** to construct prediction intervals from only sampling the query estimation module in BRANCH. Using the mean $\bar{p}_i$ and variance $\sigma_i$ of each probability query, we create lower and upper bounds:

$$p_{i-} = \begin{cases} \bar{p}_i - 2\sqrt{\sigma_i}, & \text{if } \bar{p}_i - 2\sqrt{\sigma_i} > 0, \\ \min\{p_{i1}, \ldots, p_{in}\}, & \text{otherwise.} \end{cases}$$
$$p_{i+} = \min\{\bar{p}_i + 2\sqrt{\sigma_i}, 1\}$$

These probability terms are formulated to be within $(0, 1]$, as a probability of 0 results in $k = 0$. The answers to population queries are similarly constructed, and the lower and upper bounds of all the queries for a post are combined to obtain the prediction interval $[k_{i-}, k_{i+}]$.

We run both methods 5 times on all of the user posts from Reddit in the test set. Since each user post is manually annotated with $j$ different orderings for $k_j$, we compare prediction intervals with gold standard post intervals $[\min\{k_1, \ldots, k_j\}, \max\{k_1, \ldots, k_j\}]$. We treat both intervals as integer sets for classification and evaluate with Macro-F1, recall, and precision. True positives are integers in both the gold standard and prediction intervals; false positives are integers only in the prediction interval; and false negatives are integers only in the gold standard interval. We also report model variability with the $\log_2$ size of prediction intervals. Table 9 provides the results of these prediction intervals.

**BRANCH prediction intervals result in more accurate predictions of $k$.** Re-sampling BRANCH output yields the best prediction intervals, achieving a 32.03 Macro-F1 compared to 25.60 Macro-F1 for self-consistency prediction intervals. However, re-sampling BRANCH produces much larger variance, resulting in larger prediction intervals with an average size of 11.00 compared to 3.8 for self-consistency intervals.

**Resampling Chain-of-Thought prompting is much less consistent for prediction intervals.** Resampling Chain-of-Thought also has the highest variance, with a prediction interval size of 12.70. Due to the large prediction intervals, Chain-of-Thought prompting has the highest Recall and lowest Macro-F1 at 83.02 and 24.11, respectively. Comparatively, BRANCH consistently yields better performance in $k$-estimation across multiple runs and can be utilized to create accurate prediction intervals of privacy risk.

# O   Computational Burden Analysis

We include a detailed analysis of inference time and memory usage (number of tokens and API cost) for BRANCH using GPT-4o in Table 10, which demonstrates strong performance across our tasks. For each row in the table below, we use 5 Reddit posts containing between 2 and 10 disclosures. On average, the number of LLM calls, total inference time, and tokens generated scale approximately linearly with the number of disclosures in a post.

| # of Disclosures | # of LLM Calls | Inference Time (s) | # of Tokens | API Cost ($) |
|---|---|---|---|---|
| 2 | 8 | 23 | 10,736 | 0.04 |
| 3 | 11 | 60 | 16,153 | 0.06 |
| 4 | 14 | 79 | 22,328 | 0.09 |
| 5 | 17 | 87 | 27,853 | 0.11 |
| 6 | 20 | 114 | 33,890 | 0.14 |
| 7 | 23 | 132 | 39,804 | 0.16 |
| 8 | 26 | 170 | 46,108 | 0.18 |
| 9 | 29 | 213 | 52,538 | 0.21 |
| 10 | 32 | 233 | 58,938 | 0.24 |

Table 10: Inference time and memory usage of BRANCH-GPT-4o across different disclosure counts.

# P  Licenses

We list the details and licenses for the datasets and models used in this paper.

**Datasets**

- ShareGPT: `sharegpt.com`, MIT License: `https://github.com/domeccleston/sharegpt/blob/main/license.md`

**Models**

- llama-3-8b-it-res: `https://huggingface.co/Juliushanhanhan/llama-3-8b-it-res`, Apache License v.2: `https://huggingface.co/Juliushanhanhan/llama-3-8b-it-res/tree/main`
- RoBERTa-large: `https://huggingface.co/FacebookAI/roberta-large`, MIT License: `https://huggingface.co/FacebookAI/roberta-large/tree/main`
- LLaMA-3.1-Instruct: `https://huggingface.co/meta-llama/Llama-3.1-70B-Instruct`, Llama 3.1 Community License: `https://huggingface.co/meta-llama/Llama-3.1-70B-Instruct/blob/main/LICENSE`
- LLaMA-3.3-Instruct: `https://huggingface.co/meta-llama/Llama-3.3-70B-Instruct`, Llama 3.3 Community License: `https://huggingface.co/meta-llama/Llama-3.3-70B-Instruct/blob/main/LICENSE`
- GPT-4o mini: `https://platform.openai.com/docs/models/gpt-4o-mini`, OpenAI Terms of Use: `https://openai.com/policies/terms-of-use`
- GPT-4o: `https://platform.openai.com/docs/models/gpt-4o`, OpenAI Terms of Use: `https://openai.com/policies/terms-of-use`
- o1-preview: `https://platform.openai.com/docs/models/o1-preview`, OpenAI Terms of Use: `https://openai.com/policies/terms-of-use`
- DeepSeek R1: `https://huggingface.co/deepseek-ai/DeepSeek-R1`, MIT License: `https://huggingface.co/deepseek-ai/DeepSeek-R1/blob/main/LICENSE`
- o3-mini: `https://platform.openai.com/docs/models/o3-mini`, OpenAI Terms of Use: `https://openai.com/policies/terms-of-use`

# Q  Prompts

We provide the few-shot prompts utilized for BRANCH, the baseline models, the synthetic generation of Reddit posts, and the rephrasing of Reddit text into LLaMA-3.1-Instruct text for training data for T5-style transfer below. For the baseline models, we use the same prompt for Few-Shot, Program of Thought, and Chain-of-Thought prompting but vary the reasoning steps used in the provided examples.

## Disclosure Selection Prompt

**Task**: Determine the set of disclosures that are able to be estimated online.

**Personal Disclosure**: The act of intentionally sharing personal information about oneself with another person. This information is something that others generally wouldn't know without it being shared. Examples include static personal attributes such as name, age, and gender, or events an individual engages in over time, like health and education.

**Instructions**:
1. Carefully review the post, the list of disclosure spans and their categories.
2. Refer to the subreddit name for potential information that can help search the query.
3. The subreddit name is also sometimes a disclosure, as it may indicate a location or personal attribute, such as gender. Include the subreddit as a disclosure if it is a location subreddit and it is not mentioned in **ANY** disclosure spans. Or, if the subreddit indicates a personal attribute and does not contradict with the post, include this as a valid, searchable disclosure span. A contradiction would be if the reddit user mentions explicitly another location or alternative personal attribute that goes against the norm of the subreddit.
4. Reflect deeply and thoroughly on *ALL* of the individual disclosure spans, and write down the specific/key IMPLICATIONS in a list format consistent with the post.
5. Determine which disclosure spans are able to be estimated using trustworthy online statistics and data. Examples of trustworthy surveys are surveys conducted by an official government body, university student organization, or university body. Other surveys that are trustworthy include disclosures about demographics, religion, relationships, sexual harassment, abortion, reproductive health, and mental health/psychology.
6. Include disclosures that mention **OTHER PEOPLE**. Include disclosures about locations that users **FREQUENT**, such as stores, coffee shops, etc.
7. If the listed implications can be estimated instead of the exact disclosure span itself, then include these disclosure spans. Include these disclosure spans if the implications are in these categories.
8. Present this list in <list></list>tags. EACH **Personal Disclosure** is surrounded by <answer></answer>tags, and the corresponding **DISCLOSURE CATEGORY** is surrounded by <type></type>tags.
9. If a user is planning on doing something, consider the disclosure to be valid as if the user is doing it. That is, include personal disclosure spans that have a degree of uncertainty.
10. **DO NOT** select duplicate disclosures. That is, if two disclosure spans convey the same information but are written slightly differently, **DO NOT** include both disclosures.
11. **DO NOT** change the text of the disclosure spans. Return the disclosure spans **EXACTLY AS IS**.

<examples>

**Reddit Post**: {reddit_post}
**disclosure Span**: {self_disclosure_list}
**Subreddit**: {subreddit}

**Answer**:

## Probability Ordering Prompt

**Task**: Select the orderings of disclosures to estimate sequentially, using the conditional rule of probability.

**Personal Disclosure**: The act of intentionally sharing personal information about oneself with another person. This information is something that others generally wouldn't know without it being shared. Examples include static personal attributes such as name, age, and gender, or events an individual engages in over time, like health and education.

**Instructions**:
1. Carefully review the Reddit post, the list of disclosure spans, and their categories.
2. Refer to the subreddit name for potential information that can help search the query.
3. Reflect deeply and thoroughly on *ALL* of the individual disclosure spans, and write down the potential ways to estimate each disclosure span using online data and statistics in a list format.
4. Based on your listed implications, determine an ordering to estimate the potential queries as a joint

probability. That is, given self disclosures as random variables (e.g., A = 24 years old), determine an ordering A, B, C, D, such that the joint probability can easily be decomposed into conditional probabilities P(A) * P(B | A) * P(C | B, A) * P(D | C, B, A)

5. Some guidelines for selecting the ordering are to select the disclosures that are the broadest/least specific first. For example, being a certain race or being employed is broader than disclosures about your education level or working at a specific company.

6. Select **LOCATION** disclosures first, specifically disclosures that are locations of residence first, including locations that people are moving. Locations include cities, states, countries, **universities**, or general institutions.

7. If the subreddit is an included location disclosure, select the disclosure that includes the location name **FIRST** over other location disclosures. If there are no other mentions of location disclosures, select the subreddit name as the disclosure **FIRST**. If a disclosure has the name of the subreddit within the text, **DO NOT** add a new disclosure span for the subreddit name.

8. Select disclosures that are personal attributes revolving around **ONLY** the user **FIRST**, such as **GENDER** or age or occupation. Do not order disclosures related to other people before these personal attributes, such as "relationship" or "family" disclosures.

9. For relationship disclosures, order the disclosure about the romantic partner **FIRST** (e.g., wife, girlfriend, ex, partner) before ordering disclosures about other traits in the relationship (e.g., getting a divorce, being in a abusive household, etc.). 10. Select disclosures about one's own education **BEFORE** the occupation.

11. Select disclosures based on the ease of searching the queries. For example, it is easier to search for P(male | 24 years old) than P(24 years old | male) since there are only two major categories to consider when estimating gender.

12. Order disclosures based on causality. For instance, for health disclosures, select the disclosure that may cause the health symptoms, such as an illness or particular medication.

13. Present the ordered disclosure spans **EXACTLY** as they are provided. **DO NOT** break up a disclosure span into multiple spans.

14. Present this list in <list></list>tags, with each query surrounded by <answer></answer>tags, and each category surrounded by <type></type>tags. Make sure to include ALL queries.

<examples>

Please provide the implications with deep thinking, and determine the proper ordering of disclosures.

**Reddit Post**: {reddit_post}
**disclosure Spans**: {self_disclosure_list}
**Subreddit**: {subreddit}

**Answer**:

---

## Conditional Dependencies Prompt

**Task**: Determine if the set of disclosures are conditionally independent.

**Personal Disclosure**: The act of intentionally sharing personal information about oneself with another person. This information is something that others generally wouldn't know without it being shared. Examples include static personal attributes such as name, age, and gender, or events an individual engages in over time, like health and education.

**Instructions**:
1. Carefully review the list of prior disclosure spans and their categories and the current disclosure span. Refer to the Reddit post to understand the meaning of the disclosures in context.
2. For current disclosure span X, and prior disclosure spans A, B, ..., consider the disclosure pairs (X, A), (X, B), etc.
3. Reflect deeply and thoroughly on *ALL* of the individual disclosure spans, and write down the specific/key (not general) IMPLICATIONS in a list format.
4. Based on these implications, determine if the disclosure span X can be assumed to be conditionally independent to A without drastically affecting the distribution. That is, determine if P(X | A, [Z]) = P(X | [Z]), where [Z] is a set of other random variables to condition on.
5. The prior disclosure span A is assumed to be conditionally independent if it does not drastically affect the distribution. Additionally, if removing the disclosure span as an assumption results in a probability that is MUCH easier to search up with online data and statistics, the span A is also assumed

to be conditionally independent.

6. On the other hand, a disclosure span is conditionally **DEPENDENT** if removing the conditional disclosure drastically changes the probability distribution. An example of this include gender spans when conditioned on **OCCUPATION**. For location spans, if there are other location spans that are near the current location span, then the spans are **CONDITIONALLY DEPENDENT**. Account for these implications when deciding dependence.

7. If the disclosure span is conditionally independent, then it is dropped from the conditional probability.

8. Present the conditionally dependent disclosure spans **EXACTLY** as they are provided. Do not break up a disclosure span into multiple spans. **DO NOT** include the current disclosure span in the list of prior disclosure spans.

9. Return the prior disclosureS SPANS that ARE NOT conditionally independent in <list></list>tags. Enclose the spans in <answer></answer>tags and the categories in <type></type>tags.

<examples>

Please provide the implications with deep thinking, and determine the proper conditional independence assumptions of the disclosures.

**Reddit Post**: {reddit_post}
**Prior disclosure Spans**: {prior_disclosure_list}
**disclosure Span**: {self_disclosure_span}

**Answer**:

---

## Population Query Prompt

**Task**: Generate a Google search query to determine the population of people that share this specific disclosure span.

**Personal Disclosure**: The act of intentionally sharing personal information about oneself with another person. This information is something that others generally wouldn't know without it being shared. Examples include static personal attributes such as name, age, and gender, or events an individual engages in over time, like health and education. Note: only information about the person posting (the "poster") is considered disclosure. Information about others is EXCLUDED.

**$k$-Anonymity**: Represents the number of people who possess that specific attribute or are in that specific situation. The generated search query should be aimed at understanding how many people have that particular characteristic, rather than a direct search of the span itself.

**Instructions**:
1. Carefully review the current disclosure Span(s).
2. Refer to the span category for clarity and additional information. For example, if the disclosure span is "education", this indicates that the user is a **STUDENT**.
3. Refer to the subreddit name for potential information. If the subreddit is a location, note this and use it in the query.
4. Consider the userbase of the particular subreddit. Is the subreddit used primarily by certain people, like students, children, adults? If so, utilize this specific information in your query by focusing on these people.
5. Determine if the user is a new resident of a location. If so, this number is smaller than the actual population of that location, so include this assumption as well.
6. Reflect deeply and thoroughly on *ALL* the information of the disclosure span implies for the poster, and write down the specific/key (not general) IMPLICATIONS in a list format.
7. Based on your listed implications, formulate an accurate Google search query to yield the POPULATION or NUMBER of people that share the IMPLICATIONS or ATTRIBUTES of the current disclosure span. Include the location information given. Present this query within <query></query>tags.
8. If there is no location provided for a certain population, note that the post is in English, so consider the population of interest as either the United States or the number of first-language English speakers.
9. Be specific in your query.

<examples>

Please provide the implications with deep thinking, and formulate an accurate Google search query about the poster.

**disclosure Span(s)**: {self_disclosure_list}
**Subreddit**: {subreddit}

**Answer**:

---

**Probability Query Prompt**

**Task**: Generate a Google search query to determine the $k$-Anonymity of a specific disclosure span.

**Personal Disclosure**: The act of intentionally sharing personal information about oneself with another person. This information is something that others generally wouldn't know without it being shared. Examples include static personal attributes such as name, age, and gender, or events an individual engages in over time, like health and education.

**$k$-Anonymity**: Represents the number of people who possess that specific attribute or are in that specific situation. The generated search query should be aimed at understanding how many people have that particular characteristic, rather than a direct search of the span itself.

**Instructions**:
1. Carefully review the current disclosure Span(s) about the USER/POSTER. If the current disclosure spans are separated by commas, then there are multiple current disclosure spans to estimate. Review the prior disclosure Spans, and refer to the span category for clarity and additional contextual information.
2. Refer to the subreddit name for potential information. If the subreddit is a location (i.e. a city or university), note this and use it in the query as a population of interest.
3. For all other subreddits, assume that the population of interest is the US national population or the population of English speakers as a first language. If a prior disclosure span is a subreddit name that is not a university or city, continue to use the United States as the population of interest.
4. Reflect deeply and thoroughly on *ALL* the information of the disclosure span implies for the poster, and write down the specific/key (not general) IMPLICATIONS in a list format. Implications are user traits or concrete events that are happening, not **SUBJECTIVE** thoughts, concerns, or opinions on a subject.
5. Based on your listed implications, formulate an accurate Google search query to yield the PERCENTAGE of PEOPLE sharing the prior disclosure spans THAT also share ALL the specific KEY implications of the current disclosure span. Include the location information given. Present this query within <query></query>tags.
6. If a current disclosure span consists of a comma separated list, estimate ALL the key implications of each disclosure span.
7. Note that the query includes the implications of the disclosure span rather than the exact text of the disclosure span if the implications are **EASIER** to estimate than the specific query itself.
8. Examples of these include financial situation, housing, parenthood, possession ownership, general education, health, employment status etc. These implications must be objective (i.e. something that has occurred to a user OR a user's trait).
9. For medical disclosures, create queries about the percentage of people that **HAVE** a symptom, condition, injury, etc. Do not create queries about the diagnoses.
10. The structure of the query must be **<prior disclosures>THAT <current disclosure implications>**. The disclosures that occur after the "THAT" clause are ONLY the current disclosure spans. Put the current disclosure spans after the "THAT" keyword.
11. **DO NOT** create queries in the following manner: Percentage of people <current disclosure implication(s)>**AND** <prior disclosures>.
12. If there are no prior disclosures, then the query must be formulated as **percentage of **PEOPLE** in the united states THAT <current disclosure implications >**. Do not use any other population group, you must use **PEOPLE** to refer to the general population.
13. Create queries about these disclosures objectively, as these disclosures are THINGS that happened to a user (e.g., going to an event) OR is a user trait. **DO NOT** create subjective queries about the percentage of people **HAVING AN OPINION** or **BEING CONCERNED** about something, unless **explicitly** mentioned in the span.

<examples>

Please provide the implications with deep thinking, and formulate an accurate Google search query about the poster.

**Prior disclosure Spans**: {prior_disclosure_list}
**Current disclosure Span**: {self_disclosure_span}
**Span Category**: {span_category}
**Subreddit**: {subreddit}

**Answer**:

## Query Estimation Prompt

**Task**: Estimate the value for the specific search query. Present ONLY the numeric value within <answer></answer>tags.

**Instructions**:
1. Review the search query carefully to understand what specific information is being sought.
2. Note that many queries are conditional queries. These queries are generally structured as <population of interest>THAT <specific trait/activity>. Estimate the percentage of people in the population of interest THAT share this specific trait.
3. For conditional queries, you are primarily estimating the disclosure span that occurs **AFTER** the word "THAT". Estimate the percentage of people THAT share this trait. Otherwise, estimate the number of people if it is a population query.
4. DO NOT interpret the query in the reverse direction, such as estimating the number of people sharing the specific trait that are also in the population of interest.
5. Estimate the answer based on historic data to the query. Present the numeric value within <answer></answer>tags.
6. Provide a numeric value no matter what.
7. If the number is a percentage, **REPRESENT THIS** as a decimal between 0 and 1.
8. Please give a confidence score on a scale of 0 to 1 for your prediction. The score must be in <score></score>tags.

<examples>

**Search Query**: {search_query}

Answer:

## Generalization Subquery Prompt

**Task**: Break down queries about age disclosures into multiple queries for better estimation.

**Personal Disclosure**: The act of intentionally sharing personal information about oneself with another person. This information is something that others generally wouldn't know without it being shared. Examples include static personal attributes such as name, age, and gender, or events an individual engages in over time, like health and education.

**Instructions**:
1. Review the following Google Search Query.
2. Reflect deeply and thoroughly on *ALL* of the aspects of the query and the ability to search these aspects online, and write down the specific/key (not general) IMPLICATIONS in a list format.
3. Based on your listed implications, determine if the following query can be broken down into multiple subqueries based on online statistics and data.
4. A query can be SEPARATED ONLY IF the it can be estimated with a range or generalization. Census data reports ages in ranges (e.g., 30-34), so this can be simplified by estimating the percentage of people within this age range. The second query should ask for the number of years in that age range, such that dividing the percentage of people within this age range by the number of years in the age range provides a uniform estimate for the percentage of people in a single year.
5. Additionally, relationship status queries mention a gender of a potential partner (e.g., a wife, a girlfriend, your fiancee), which can be broken down into **TWO** subqueries that divides the gender of the partner. This can be done by creating the first query for broad relationship or family statistics.

For example, the disclosure "wife" can be generalized to the percentage of people that are married. Then, specify the number of bins to divide this by. You can estimate this with the expected number of genders. Dividing the query by this number will provide an estimate to the percentage of people that are married to a specific gender, i.e. a wife.

6. If the query cannot be estimated by broken down into subqueries, **RETURN** the original query in <query></query>tags.

7. Otherwise, return the new queries in <list></list>tags. Determine the mathematical operation needed to be used to combine the subqueries into the original query, and return this in <math></math>tags.

<examples>

Please provide the implications with deep thinking, and determine if the query needs to be broken down into subqueries.

**Query**: {query}

**Answer**:

---

## Discrete Subquery Prompt

**Task**: Determine if disclosures can be broken down into multiple queries for better estimation.

**Personal Disclosure**: The act of intentionally sharing personal information about oneself with another person. This information is something that others generally wouldn't know without it being shared. Examples include static personal attributes such as name, age, and gender, or events an individual engages in over time, like health and education.

**Instructions**:
1. Review the following Google Search Query.
2. Reflect deeply and thoroughly on *ALL* of the aspects of the query and the ability to search these aspects online, and write down the specific/key (not general) IMPLICATIONS in a list format.
3. Determine if the following queries CONTAINS TWO SEPARATE, DISCRETE disclosure spans in the CONDITIONAL CLAUSE (e.g., "THAT are CS or Economics", etc.).
4. Discrete disclosure spans in the conditional clause usually include the mention of the word "OR", where one disclosure is true or the other is true. If both or multiple are valid, then this is not a discrete disclosure span.
5. There is a **CONDITIONAL CLAUSE** with the word "THAT". The presence of the word "OR" **MUST BE INSIDE** the conditional clause. That is, a discrete query that can be broken down will have the structure <... "THAT" ... OR ...>. Determine what the conditional clause is, and then determine if the word "OR" is in the **CONDITIONAL CLAUSE**. If it is, then the queries can be broken down.
6. If the word "or" comes **BEFORE** the conditional clause, **DO NOT** separate the query into multiple subqueries. The conditional clause **MUST CONTAIN** the word **OR**.
7. **DO NOT BREAKDOWN** disclosure spans that are just lists separated by commas. These do not indicate that there is a hidden "OR" in the span.
8. If there are two discrete spans in the conditional clause, break down the query into **TWO** subqueries that can be added up to estimate the original query.
9. If the query **DOES NOT** have discrete spans, then it cannot be estimated by breaking down the query. **RETURN** the original query in <list></list>tags.
10. Otherwise, return the new queries in a list with <list></list>tags. Surround each query in <answer></answer>tags. Determine the mathematical operation needed to be used to combine the subqueries into the original query, and return this in <math></math>tags.

<examples>

Please provide the implications with deep thinking, and determine if the query needs to be broken down into subqueries.
**Query**: {query}

**Answer**:

## Evaluate Answer Prompt

**Task**: You are an expert on recognizing incorrect information. Determine if the answer to a query is roughly correct.

**Personal Disclosure**: The act of intentionally sharing personal information about oneself with another person. This information is something that others generally wouldn't know without it being shared. Examples include static personal attributes such as name, age, and gender, or events an individual engages in over time, like health and education.

**Instructions**:
1. Review the Google Search Query and the answer.
2. This answer is produced by an arbitrary large language model, and it is not certain that the answer is accurate.
3. Using historic statistics and analogous data, determine if the provided answer to the query is correct or not.
4. Provide your reasoning as to why the answer is likely to be correct or not.
5. Return your decision, "Yes" or "No" in <answer></answer>tags based on if the answer trustworthy. Only provide "No" if it is very likely that the estimate is incorrect.

<examples>

Explain your thoughts.
**Query**: {query}
**Answer**: {answer}

**Evaluation**:

## Simplify Query Prompt

**Task**: Simplify the following Google Search query about disclosures.

**Personal Disclosure**: The act of intentionally sharing personal information about oneself with another person. This information is something that others generally wouldn't know without it being shared. Examples include static personal attributes such as name, age, and gender, or events an individual engages in over time, like health and education.

**Instructions**:
1. Review the following Google Search Query.
2. Reflect deeply and thoroughly on *ALL* of the aspects of the query and the ability to search these aspects online, and write down the specific/key (not general) IMPLICATIONS in a list format.
3. Based on your listed implications, determine what edits can be made that does not severely change the query but make it easier to estimate with online statistics and data. Examples of these changes are to change the location of interest (e.g., from a city to the country) or to make a generalization of the disclosure.
4. Return the simplified query in <query></query>tags.

<examples>
Please provide the implications with deep thinking, and formulate an simplified Google search query about the poster.

**Query**: {query}

**Answer**:

## Few-Shot; Chain-of-Thought; and Program of Thought Prompts

**Task**: Estimate a k-number corresponding to a list of personal disclosures and the Reddit post.

**Personal Disclosure**: The act of intentionally sharing personal information about oneself with another person. This information is something that others generally wouldn't know without it being shared. Examples include static personal attributes such as name, age, and gender, or events an

individual engages in over time, like health and education. Note: only information about the person posting (the "poster") is considered disclosure. Information about others is EXCLUDED.

**$k$-Anonymity**: Represents the number of people who possess that specific attribute or are in that specific situation. The generated search query should be aimed at understanding how many people have that particular characteristic, rather than a direct search of the span itself.

**Instructions**:
1. Carefully review the Reddit post and disclosure Span(s).
2. Refer to the span category for clarity and additional information. For example, if the disclosure span is "education", this indicates that the user is a **STUDENT**.
3. Refer to the subreddit name for potential information. If the subreddit is a location, note this and use it in the query.
4. Reflect deeply and thoroughly on *ALL* the information of the disclosure span implies for the poster, and write down the specific/key (not general) IMPLICATIONS in a list format.
5. Based on your listed implications, determine which disclosures can be estimated with online statistics or trustworthy surveys.
6. Based on the estimatable disclosures, estimate the $k$-Anonymity value, which is a number $k$ that represents all the people that share the specific disclosure spans out of everyone in the world.
7. If there is a location mentioned in the post, use that as the initial starting population. Otherwise, estimate the population as the number of native English speakers (around 500 million) people.
8. Present this k-value as an exact integer number within <answer></answer>tags.

<examples>

**Reddit Post**: {reddit_post}
**Example disclosure Span**: {self_disclosure_list}
**Example Subreddit**: {subreddit}

**Example Answer**:

---

## Synthetic Generation Prompt

You are a {age} {gender} person at {location}. You are {race}. You are a {occupation}. You are {income_level}. You are {orientation} and currently {relationship}. You currently feel {emotional_condition}. You have {events} soon.

You will be given example Reddit posts from a subreddit as reference. You are tasked with creating a new Reddit post that is **SIMILAR** to the provided posts in writing style and content.

Your response should reflect your profile, should be specific, and include at least **4** of your personal **distinct** elements in the response. Do not be vague or generic.

The personal elements must occur **naturally** in the posts similar to the example posts provided. Additionally, once using a personal element, **DO NOT** repeat the information again. For relationship elements, do not center the post or the paragraph on the relationship.

Write your answer like a Reddit comment or post. DO NOT MENTION the words "Reddit", "fellow", or "Redditor" in your response. Return your new post in <answer></answer>tags.

Write your responses with similar formulations and use of language as the assistant in the above examples. Notice that capitalization, punctuation, and correct grammar are often neglected, reflecting a relaxed, colloquial style.

Provided Examples: <examples>

Subreddit: {subreddit}

New Post:

