# OpenReview forum: "Probabilistic Reasoning with LLMs for Privacy Risk Estimation"
_NeurIPS.cc/2025/Conference — NeurIPS 2025 poster_

### Official Review · Reviewer_nErv · 2025-06-30

**Clarity:** 3
**Significance:** 3
**Originality:** 3
**Rating:** 5
**Confidence:** 4

**Summary:**

This paper introduces BRANCH (Bayesian Network Reasoning for k-ANonymity using Conditional Hierarchies), a novel methodology for estimating privacy risk in user-generated content. The work addresses the challenge of quantifying how many people worldwide share the personal attributes disclosed in a text, providing users with an interpretable k-value representing their privacy risk. BRANCH factorises joint probability distributions of personal information by constructing Bayesian networks, estimating individual attribute probabilities separately, and combining them to compute the final k-value. The methodology achieves 72.61% accuracy in k-value estimation, representing a 13% improvement over o3-mini with chain-of-thought reasoning. The authors also demonstrate that LLM uncertainty correlates with prediction accuracy, with high-variance predictions being 37.47% less accurate on average.

**Questions:**

1. How does the computational cost of BRANCH scale with the number of disclosures? Could you provide a detailed analysis of inference time and memory requirements? What strategies could be employed to improve efficiency whilst maintaining accuracy?
2. Threat Model Generalisation: Could the methodology be extended to handle partial knowledge threat models where adversaries only have access to subsets of disclosed information? How would this affect the Bayesian network structure and probability estimation?
3. Model Selection and Hyperparameter Justification: The paper would benefit from clearer justification for specific model choices (e.g., why GPT-4o and o3-mini specifically) and hyperparameter selections (e.g., confidence threshold of 0.55, temperature of 0.7, 5 reruns for uncertainty estimation).
4. Generalisation Beyond Current Dataset: How would BRANCH perform on documents with significantly more disclosures (>10) or in domains beyond Reddit and ShareGPT? What adaptations would be necessary for other types of sensitive content or different cultural contexts?

**Ethical Concerns:**

["NO or VERY MINOR ethics concerns only"]

**Final Justification:**

The author strongly addressed my concerns in the paper, and this would be a good addition to the conference proceedings.

**Limitations:**

Yes

**Paper Formatting Concerns:**

The paper generally follows NeurIPS formatting guidelines

**Quality:**

3

**Strengths And Weaknesses:**

## Strengths
- Quality: The paper presents a methodologically sound approach to privacy risk assessment. The Bayesian network framework is well-motivated and appropriately designed for decomposing complex joint probabilities. The experimental design is comprehensive, including proper statistical significance testing, ablation studies, and error analysis. The evaluation covers multiple domains (Reddit posts and ShareGPT conversations) and includes uncertainty quantification.
- Clarity: The paper is generally well-written with clear exposition of the methodology. Figure 2 effectively illustrates the BRANCH framework, and the examples in Figure 1 clearly demonstrate the limitations of chain-of-thought approaches. The mathematical formulation is precise, and the experimental setup is described in sufficient detail for reproducibility.
- Significance: This work addresses an important practical problem in privacy protection and introduces a novel application of probabilistic reasoning to LLMs. The k-anonymity framework provides users with interpretable risk estimates, which could have real-world applications in privacy-aware systems. The methodology demonstrates meaningful improvements over existing approaches and could inform future research in privacy-preserving technologies.
- Originality: The application of Bayesian networks to privacy risk estimation with LLMs is novel. The decomposition of joint probabilities into conditional dependencies represents a creative approach to handling complex attribute interactions. The integration of uncertainty quantification and the development of a comprehensive dataset for this task contribute to the work's originality.
## Weaknesses
- Scalability and Computational Burden: The methodology requires multiple LLM inference steps for Bayesian network construction, query generation, and individual probability estimation. This creates significant computational overhead, particularly when incorporating uncertainty quantification. The scalability to documents with more than 10 disclosures remains unexplored.
- Limited Threat Model: The current approach assumes adversaries have complete knowledge of disclosed attributes, which may not reflect realistic threat scenarios. The lack of specialised threat models where users can specify what information adversaries possess limits practical applicability.
- Dataset Limitations: The maximum number of disclosures per document is around 10, and the dataset is primarily focused on Reddit and ShareGPT conversations. The generalisability to other domains and longer documents with more complex attribute interactions is unclear.
- Ground Truth Validation: Whilst the authors report 22.24% average percentage error compared to census data, there are concerns about compounding errors when combining multiple probability estimates. The reliance on human annotators for ground truth introduces potential biases.

---

> ### Author Rebuttal · Authors · 2025-07-31
>
> >Scalability and Computational Burden: The methodology requires multiple LLM inference steps for Bayesian network construction, query generation, and individual probability estimation.
>
> We thank the reviewer for their comments regarding the computational burden of BRANCH. Similar to other advanced reasoning models such as o3, BRANCH incurs higher latency with the trade-off of having better accuracy. The focus of this work is to develop a framework that can utilize LLM’s probabilistic reasoning capabilities to estimate privacy risk as accurately as possible. As such, BRANCH is an advanced reasoning model with multiple LLM inference steps that prioritizes interpretability and robustness over inference time speed.
>
> However, we acknowledge that the multi-step inference pipeline of BRANCH poses a practical limitation. We will clarify this in the revised paper and provide more concrete discussion of future research directions to mitigate this limitation, including:
>
> - *LLM Output Caching*: By caching intermediate outputs, especially for frequently seen or structurally similar inputs, we can significantly reduce the computation time.
> - *Lightweight Model Substitution*: In many cases, such as simpler posts, we can utilize Chain-of-Thought outputs or lightweight, distilled models to approximate certain stages of the BRANCH pipeline without full LLM calls.
> - *Pipeline Optimization*: We are investigating batching and combining BRANCH steps to reduce the total number of LLM calls required to estimate privacy risk.
>
> >Dataset Limitations: The maximum number of disclosures per document is around 10, and the dataset is primarily focused on Reddit and ShareGPT conversations. The generalisability to other domains and longer documents with more complex attribute interactions is unclear. What adaptations would be necessary for other types of sensitive content or different cultural contexts?
>
> Due to the cost of annotating privacy risk in text, we intentionally focused on the natural distribution of atomic user posts on domains rich in personal attributes that map directly to structured and publicly available ground truth sources (e.g., census data, university records). This required significant manual effort to ensure reliable gold-standard privacy risk estimates. Notably, the distribution of disclosures in our dataset is usually between 3–6 per post and rarely extends past 8 disclosures, which closely matches the natural distribution of personal information across various subreddits, including longer Reddit posts on communities like r/AITA. In these domains, we demonstrate that BRANCH is well-equipped to predict the privacy risk of sensitive documents, which serves as a strong foundation for initial development. We do not anticipate any fundamental limitations that would prevent BRANCH from generalizing to texts with more than 10 disclosures, especially since our analysis indicates that BRANCH outperforms CoT-based methods in handling dense attribute interactions as the number of disclosures increases. However, we have not extensively tested this case, as it is rare in naturally occurring data. We plan to explore such settings in future work, including domains with more complex attribute dependencies, especially where adversarial knowledge varies in granularity or source.
>
> In culturally or contextually specific settings (e.g., financial communities with company-specific norms), additional adaptations will be necessary. If domain-specific ground truth data is available, we can incorporate retrieval-augmented generation (RAG) modules to contextualize inference. In lower-resource settings, fine-tuning expert estimators or adapting BRANCH with weak supervision from auxiliary statistics may improve reliability.
>
> >Limited Threat Model: The current approach assumes adversaries have complete knowledge of disclosed attributes, which may not reflect realistic threat scenarios. The lack of specialised threat models where users can specify what information adversaries possess limits practical applicability.
>
> We thank the reviewer for highlighting the point regarding the generalizability of the threat model and agree that this is a promising research direction for the future. In this work, we primarily introduce an unexplored framework for estimating privacy risk using probabilistic reasoning with LLMs, focusing on helping users determine if an individual person could identify them from their text. To ground our methodology and establish an upper bound on potential risk, we focus on a strong adversary with access to knowledge of all post-specific disclosure contexts. This setup enables a rigorous evaluation of the maximum possible privacy leakage under realistic but constrained assumptions.
>
> We fully agree that extending BRANCH to accommodate more personalized threat models will further enhance its applicability. Modeling such adversaries, however, introduces new challenges, as it requires inferring both user preferences and adversary-specific background knowledge and uncertainty. Future work could study user-personalized threat modeling, where privacy risk is estimated relative to the knowledge scope of specific, personalized adversaries. This would include adversaries with access to a user’s post history, scenarios where attacker knowledge is drawn from only public data, and threat models that have access to machine learning tools, similar to existing work in authorship attribution [1]. To accommodate these threat models, BRANCH would need to condition the Bayesian network on only the known subset, which may alter its structure, conditional dependencies, and downstream probability estimates. We will update the limitations section of our paper to integrate this discussion for more nuanced and personalized privacy assessments in future research.
>
> [1] Huang, B., Chen, C., & Shu, K. (2024). Authorship Attribution in the Era of LLMs: Problems, Methodologies, and Challenges. ACM SIGKDD Explorations Newsletter, 26, 21 - 43.
>
> >Ground Truth Validation: Whilst the authors report 22.24% average percentage error compared to census data, there are concerns about compounding errors when combining multiple probability estimates. The reliance on human annotators for ground truth introduces potential biases.
>
> We reduce the risk of annotator bias through verifying our dataset with double annotation (see section 4.1) and achieving a high Spearman correlation (ρ = 0.916), which indicates strong consistency among annotators with relevant expertise. Additionally, the reported 22.24% average percentage error is calculated over population groups of size ≥4, which reflects the typical populations covered in our dataset. Thus, while compounded errors are a theoretical concern, their practical impact remains low under our annotation protocol, especially for the domains (e.g., Reddit, ShareGPT) that utilize census information or public databases. We will update this discussion in the paper.
>
> >How does the computational cost of BRANCH scale with the number of disclosures? Could you provide a detailed analysis of inference time and memory requirements? What strategies could be employed to improve efficiency whilst maintaining accuracy?
>
> We thank the reviewer for raising this important question. We include a detailed analysis of inference time and memory usage for BRANCH using GPT-4o, which demonstrates strong performance across our tasks. For each row in the table below, we use 5 Reddit posts containing between 2 and 10 disclosures. On average, the number of LLM calls, total inference time, and tokens generated scale approximately linearly with the number of disclosures in a post.
>
> To improve efficiency while maintaining accuracy, we are actively exploring several strategies:
> - *LLM Output Caching*: Caching intermediate outputs, particularly for frequently occurring or structurally similar disclosures, can substantially reduce redundant computation.
> - *Lightweight Model Substitution*: For simpler posts or lower-risk scenarios, we can substitute specific stages of BRANCH with Chain-of-Thought prompting or distilled lightweight models, reducing the number of full LLM calls required.
> - *Pipeline Optimization*: We are optimizing the pipeline through batching and combining steps where possible, further decreasing the total number of model queries needed for inference.
>
>
> We will include the analysis table and a discussion of these strategies in the final version of the paper to better contextualize the computational tradeoffs of BRANCH.
> | # of Disclosures | # of LLM Calls | Inference Time (s) | # of Tokens | API Cost ($) |
> |--|--|--|--|--|
> | 2 | 8  | 23  | 10,736  | 0.04  |
> | 3 | 11  | 60  | 16,153  | 0.06  |
> | 4 | 14  | 79  | 22,328  | 0.09  |
> | 5 | 17  | 87  | 27,853  | 0.11  |
> | 6 | 20 | 114 | 33,890 | 0.14 |
> | 7 | 23 | 132 | 39,804 | 0.16 |
> | 8 | 26 | 170 | 46,108 | 0.18 |
> | 9 | 29 | 213 | 52,538 | 0.21 |
> | 10   | 32 | 233 | 58,938 | 0.24 |
>
> >Model Selection and Hyperparameter Justification: The paper would benefit from clearer justification for specific model choices (e.g., why GPT-4o and o3-mini specifically) and hyperparameter selections (e.g., confidence threshold of 0.55, temperature of 0.7, 5 reruns for uncertainty estimation).
>
> We thank the reviewer for their suggestions. We selected GPT-4o and o3-mini as models for BRANCH due to their strong baseline performances, and hyperparameters were tuned using the development set of our dataset: a confidence threshold of 0.55 provided the highest range accuracy and prevented the model from over-simplifying or under-simplifying queries; temperature values of 0 and 1 were tested but underperformed compared to 0.7. Finally, we used 5 reruns for uncertainty estimation as a practical compromise, balancing robustness with the computational cost of BRANCH’s multi-step pipeline. We will clarify all of these hyperparameter choices in the paper.

---

> > ### Comment · Reviewer_nErv · 2025-08-06
> >
> > The rebuttal addresses most concerns well and shows the authors have a clear understanding of their work's limitations and future directions. The computational analysis table is particularly valuable. However, the ground truth validation response could do with strengthening. Overall The strong responses on scalability and the concrete computational analysis could move this to an accept. I have updated my score.

---

### Official Review · Reviewer_Pbz9 · 2025-07-02

**Clarity:** 3
**Significance:** 3
**Originality:** 3
**Rating:** 5
**Confidence:** 2

**Summary:**

This paper introduces BRANCH, a novel framework that enables large language models to estimate privacy risk in user-generated text by modeling the joint distribution of self-disclosed attributes via Bayesian networks. It achieves significantly better performance than existing reasoning baselines and provides interpretable k-anonymity scores along with uncertainty estimates.

**Questions:**

Please refer to Weaknesses.

**Ethical Concerns:**

["NO or VERY MINOR ethics concerns only"]

**Limitations:**

Yes

**Quality:**

3

**Strengths And Weaknesses:**

Strengths

• Novel LLM task and framework: The paper introduces a new task of text-level privacy risk estimation using LLMs and proposes BRANCH. This framework models the joint distribution of personal attributes via Bayesian networks. The formulation is original and practically relevant.

• Well-structured and interpretable methodology: BRANCH decomposes complex probabilistic reasoning into conditional probability estimation and recombination. The framework is logically rigorous, clearly structured, and leverages LLMs’ natural language querying capabilities to produce interpretable outputs.

• Comprehensive experiments and superior performance: The authors conduct thorough evaluations across multiple competitive baselines. Results consistently demonstrate the validity and superiority of BRANCH.

Weaknesses

• The paper does not analyze how consistent the LLM-generated Bayesian network structures are across multiple runs. It is unclear whether the inferred dependency orderings and structures are consistent, especially when statistical priors are noisy or unavailable.

•  BRANCH assumes certain variables to be conditionally independent (e.g., gender and housing status) to simplify modeling. However, the paper does not quantify the impact of incorrect independence assumptions on the final k-value estimation.

• The approach relies heavily on LLM-estimated demographic statistics (e.g., tech employment rates, birth rates), but the paper does not include a formal sensitivity analysis to assess how deviations in these estimates affect the final privacy risk predictions.

• While the model is evaluated on Reddit and ShareGPT posts, a substantial portion of the training data is synthetic. It is uncertain how well BRANCH generalizes to diverse, real-world scenarios, especially outside the synthetic training distribution.

---

> ### Author Rebuttal · Authors · 2025-07-31
>
> >The paper does not analyze how consistent the LLM-generated Bayesian network structures are across multiple runs. It is unclear whether the inferred dependency orderings and structures are consistent, especially when statistical priors are noisy or unavailable.
>
> We thank the reviewer for pointing out the importance of evaluating the consistency of LLM-generated Bayesian network structures. In our paper, we conducted 5 runs (see lines 333-334) of the full BRANCH pipeline and found that the overall performance of the BRANCH model remained stable, suggesting robustness in Bayesian network elicitation.
>
> To further quantify structural consistency, we analyzed the topological orderings of the Bayesian network using Kendall’s Tau across the 5 runs and obtained a score of 0.682, indicating a strong correlation in variable orderings across runs. Additionally, we computed the mean structural Hamming distance (SHD) between the induced shared subgraphs across different BRANCH runs. The mean SHD was 1.29, with a minimum of 0 and a maximum of 10, showing that only ~1.3 edge operations (additions, deletions, or reversals) are typically needed to transform one network into another across the 5 BRANCH runs. For context, the mean maximum possible SHD (i.e., comparing a fully connected subgraph to an entirely disconnected one) is 11.36, further affirming that the Bayesian network structures are highly consistent across runs despite noisy or missing priors. These findings collectively suggest that the BRANCH pipeline produces stable and consistent Bayesian network structures, even under variability in LLM outputs.
>
> >BRANCH assumes certain variables to be conditionally independent (e.g., gender and housing status) to simplify modeling. However, the paper does not quantify the impact of incorrect independence assumptions on the final k-value estimation.
>
> We appreciate the reviewer’s point about the impact of independence assumptions. We conducted ablation studies (included in Appendix J) to compare different conditional independence assumptions (i.e., completely dependent structure or completely independent structure). We find that BRANCH with our selected conditional dependence assumptions consistently achieves the best performance, indicating that these modeling choices are empirically justified. We will add this discussion to the main paper.
>
> To further assess the impact of incorrect independence assumptions, we conducted a detailed error analysis of the best-performing BRANCH system (included in Appendix K). We find that approximately 40% of the observed errors are due to issues in query generation, many of which can be traced to overly strong conditional independence assumptions. This suggests that while BRANCH is robust overall, a portion of its errors stem from independence assumptions, motivating future work on more accurate data-driven modeling of these dependencies.
>
> >The approach relies heavily on LLM-estimated demographic statistics (e.g., tech employment rates, birth rates), but the paper does not include a formal sensitivity analysis to assess how deviations in these estimates affect the final privacy risk predictions.
>
> We thank the reviewer for this point. While we do not conduct a full formal sensitivity analysis, we explore the variability of the final k-values under self-consistency sampling for individual LLM-estimated demographic statistics (see Section 7). Specifically, we obtain multiple answers for the same queries and find that the resulting k-values constructed from these individual query answers exhibit low variance across samples, suggesting that the pipeline is robust to fluctuations in LLM-estimated demographic statistics. This indicates that the LLM is internally consistent when estimating these quantities in the domains in our dataset and that such variability does not significantly affect the final privacy risk predictions. We also empirically validate their reliability by comparing LLM-based estimates of individual queries to human-annotated ground truth statistical estimates in Section 5.2, observing that LLMs are quite accurate at producing statistical distributions in these domains. We will further clarify this discussion in the paper.
>
> >While the model is evaluated on Reddit and ShareGPT posts, a substantial portion of the training data is synthetic. It is uncertain how well BRANCH generalizes to diverse, real-world scenarios, especially outside the synthetic training distribution.
>
> We appreciate the reviewer’s concern regarding generalization beyond our current domains. In this work, we primarily introduce an unexplored framework for estimating privacy risk using probabilistic reasoning with LLMs, focusing on helping users determine if an individual person could identify them from their text. BRANCH is evaluated solely on real-world user-written Reddit and ShareGPT posts, both of which are common platforms for sharing personal disclosures. That said, we acknowledge that these domains may not capture the full diversity of real-world scenarios.
>
> Future work could look at developing user studies to evaluate BRANCH in more real-world contexts and exploring additional threat models and contexts (e.g., an individual who also knows additional information not present in the post) to better understand BRANCH’s robustness across use cases. Cross-domain generalization is also an important direction, including applications to Amazon reviews, job forums, and other online platforms where personal information is frequently shared. These efforts will help investigate how well BRANCH can generalize to diverse, real-world scenarios.

---

### Official Review · Reviewer_9q4K · 2025-07-02

**Clarity:** 3
**Significance:** 3
**Originality:** 4
**Rating:** 5
**Confidence:** 4

**Summary:**

This paper introduces BRANCH (Bayesian Network Reasoning for k-ANonymity using Conditional Hierarchies), a novel Large Language Model (LLM) methodology to estimate the k-privacy value of user-generated text. The k-privacy value represents the size of the population matching disclosed personal information, aiming to provide end-users with an interpretable privacy risk estimate without requiring access to comprehensive databases. The authors also present a new dataset for privacy risk estimation, consisting of user-LLM conversations and Reddit posts.

Summary:

The core problem addressed is the difficulty for end-users to understand the privacy implications of disclosing personal information online. The paper proposes BRANCH, which tackles this by:

- Self-Disclosure Identification: Identifying personal disclosures in user documents.

- Bayesian Network Structure Elicitation: Implicitly constructing a Bayesian network to capture interdependencies between attributes by ordering disclosures and determining conditional dependencies.

- Query Generation and Subquery Decomposition: Converting conditional probabilities into search queries, which can be decomposed into subqueries for better estimation.

- Query Estimation and Probability Recombination: Using LLMs' inherent knowledge (potentially augmented with Retrieval-Augmented Generation) to estimate query answers and then recombining these probabilities using a Python interpreter to calculate the final k-value.

- Experiments show BRANCH estimates the k-value correctly 73% of the time, a 13% improvement over 03-mini with Chain-of-Thought reasoning. The paper also highlights that LLM uncertainty correlates with accuracy.

**Questions:**

- Are there any error modes where BRANCH performs worse than CoT? Some examples of when BRANCH fails would be valuable to understand its limitations.

- Why not use ground-truth census APIs or structured data sources to improve query answers? The model currently relies on LLMs’ internal knowledge, which may drift. Could retrieval-augmented models improve robustness?

- How was the “gold” k value validated in cases where no ground truth exists?

- For many Reddit or ShareGPT examples, ground truth is unclear. What confidence do we have in those values?

- What criteria were used to select the “ideal” ordering of attributes in the Bayesian network? The paper says LLMs are instructed to find a “feasible ordering” — but is there a heuristic or constraint guiding that process?

**Ethical Concerns:**

["NO or VERY MINOR ethics concerns only"]

**Final Justification:**

The authors have answered all my concerns in a very detailed manner and their justification makes sense to me.

**Limitations:**

- The attacker model assumes someone familiar with the document context and matching on all attributes, but ignores auxiliary knowledge, linkage attacks, or adversaries with machine learning tools. The assumption that k provides meaningful protection without database access is not well-defended against modern adversaries.

- The framework assumes accurate estimates from LLMs for subqueries, but no adversarial robustness or sensitivity analysis is performed (e.g., effect of minor wording changes or noisy queries).

- As mentioned earlier, a significant portion of the training data (50/220) is synthetic, generated by LLaMA. There’s no ablation to test the effect of synthetic data on generalization, which could affect conclusions.

- The BRANCH framework depends on manually annotated disclosures, Bayesian network structures, and conditional dependencies. This raises scalability and reproducibility concerns — it’s unclear how well the pipeline performs in the absence of such annotations.

**Paper Formatting Concerns:**

Figure 3 (scatter plots of predicted vs ground truth k) is critical, but its axis labels and color usage could be more accessible and publication-ready. My suggestion is to separate explanatory text from figures and ensure all figures have standalone captions that summarize their core message.

**Quality:**

3

**Strengths And Weaknesses:**

Strengths:

- Novel Problem Framing: The paper shifts the focus of privacy risk assessment from dataset owners to end-users, providing an interpretable k value. This user-centric approach is highly relevant and practical for online privacy.



- Innovative Methodology (BRANCH): Decomposing the complex problem of probabilistic reasoning into manageable steps (Bayesian network construction, query generation, subquery decomposition, and probability recombination) is a strong technical contribution. The use of LLMs to implicitly construct Bayesian networks and estimate conditional probabilities is a creative application of their reasoning capabilities.



- New Dataset and Evaluation: The creation of a new, manually annotated dataset of Reddit posts and user-LLM conversations specifically for privacy risk estimation is valuable. The rigorous annotation process, including double annotation and ground truth validation against census records, indicates a high-quality dataset.





- Interpretability: The k-anonymity metric provides an intuitive and interpretable value for users, which is crucial for practical privacy tools.


- Thorough Error Analysis: The detailed analysis of Chain-of-Thought errors (conditional dependency, lack of subqueries, PII) and BRANCH errors provides valuable insights into the challenges of probabilistic reasoning for LLMs in this domain.


Weaknesses:

- Computational Burden: The paper acknowledges that BRANCH can pose a computational burden due to multiple LLM inference steps. While caching LLM outputs is suggested for future work, this is a significant practical limitation for real-time applications.

- Generalizability of Threat Model: The current threat model assumes an adversary with knowledge of post-specific disclosure contexts. While practical, exploring more generalized threat models where adversaries have varying levels of access to information (e.g., publicly available databases) would strengthen the work.

- Lack of User Study: The paper states that no specific user study has been conducted to test the model for providing quantifiable privacy risk estimates, despite prior HCI studies motivating the work. This is a missed opportunity to validate the real-world utility and user perception of the tool. There’s no user study or even simulated deployment experiment to validate how users interpret or benefit from k.

- Potential Biases: The paper identifies that it has not analyzed whether the model itself may have biases that impact its use for certain demographics. This is a critical ethical consideration, especially for marginalized or vulnerable populations, and should be addressed before deployment.


- Dataset Size and Scope: While the dataset is well-constructed, 220 documents (180 Reddit, 40 ShareGPT) might be considered modest for training robust LLM-based reasoning systems, especially given the complexity of probabilistic reasoning. The maximum of 10 disclosures per post is also a limitation for more extensive personal profiles. 50/220 documents are synthetic, and LLaMA-generated data may carry stylistic or topical biases. While synthetic data is common for privacy work, no ablation is presented on model performance with vs. without these documents. This raises concerns about data contamination or optimism bias.
Another thing I am a little worried about is that while BRANCH outperforms CoT in range accuracy, many documents only require reasoning over 2–3 attributes. For these simpler cases, CoT performs comparably, and the performance gap emerges only for 4+ attributes (Figure 4).
This limits the claim that BRANCH is broadly superior; it is specifically better for complex inputs, not necessarily the dominant approach in all privacy scenarios.

- The “privacy strength meter” analogy is appealing, but there's no evidence this would aid users in practice.

- The methodology relies heavily on manual annotation of disclosures, Bayesian structures, and equations, which limits scalability and reproducibility. While inter-annotator agreement is reported (ρ = 0.916), constructing Bayesian networks with conditional dependencies is subjective and nontrivial.

---

> ### Author Rebuttal · Authors · 2025-07-31
>
> >Computational Burden: The paper acknowledges that BRANCH can pose a computational burden due to multiple LLM inference steps.
>
> We thank the reviewer for their comments regarding the computational burden of BRANCH. Similar to other advanced reasoning models such as o3, BRANCH incurs higher latency with the trade-off of having better accuracy. The focus of this work is to develop a framework that can utilize LLMs’ probabilistic reasoning capabilities to estimate privacy risk as accurately as possible. As such, BRANCH is composed of multiple LLM inference steps that prioritize interpretability and robustness over inference time speed.
>
> However, we acknowledge that the multi-step inference pipeline of BRANCH poses a practical limitation. We will provide more concrete discussions of future research directions to mitigate this limitation, including:
>
> - LLM Output Caching.
> - Lightweight Model Substitution. In simpler posts, lightweight, distilled models can approximate BRANCH without full LLM calls.
> - Pipeline Optimization to combine BRANCH steps.
>
> >Generalizability of Threat Model: The current threat model assumes an adversary with knowledge of post-specific disclosure contexts. The attacker model ignores auxiliary knowledge, linkage attacks, or adversaries with machine learning tools.
>
> We thank the reviewer for highlighting the point regarding the generalizability of the threat model and agree that this is a promising research direction for the future. In this work, we primarily introduce an unexplored framework for estimating privacy risk using probabilistic reasoning with LLMs, focusing on helping users determine if an individual person could identify them from their text. To ground our methodology and establish an upper bound on potential risk, we focus on a strong adversary with access to knowledge of all post-specific disclosure contexts. This setup enables a rigorous evaluation of the maximum possible privacy leakage under realistic but constrained assumptions.
>
> We fully agree that extending BRANCH to accommodate more personalized threat models will further enhance its applicability. Modeling such adversaries, however, introduces new challenges, as it requires inferring both user preferences and adversary-specific background knowledge and uncertainty. Future work could study user-personalized threat modeling, where privacy risk is estimated relative to the knowledge scope of specific, personalized adversaries. Our current work defines privacy risk depending on the specific threat model. As such, our privacy risk value can change and be used to measure privacy risk against modern adversaries. We will update the limitations section to integrate this discussion for personalized privacy assessments.
> >Lack of User Study: The paper states that no specific user study has been conducted to test the model for providing quantifiable privacy risk estimates. There’s no user study or even simulated deployment experiment to validate how users interpret or benefit from k.
>
> We acknowledge the reviewer’s concern regarding the lack of a user study in the current version of the work. Our primary focus has been to introduce a new framework and evaluation dataset for estimating the privacy risk of sensitive posts on online forums using probabilistic reasoning with LLMs. Our contributions center on technical feasibility and the design of interpretable, post-specific privacy risk signals.
>
> We agree that understanding how users interpret and benefit from BRANCH’s privacy risk value is essential for practical deployment. While we are in the process of conducting such a study, user studies require careful design and extended timelines, and these studies are outside the scope of the current paper. Our goal in this work is to establish a robust, generalizable framework for privacy risk estimation, independent of application-specific deployments. We believe that a dedicated HCI-focused follow-up would be better suited to fully explore user-facing aspects, including presentation modes (e.g., privacy risk meter) and how users adjust behavior in response.
> >Potential Biases: The paper identifies that it has not analyzed whether the model itself may have biases that impact its use for certain demographics.
>
> We appreciate the reviewer for raising this important point. Our work presents a methodological contribution for evaluating user-side privacy using LLMs, not a deployable system intended for real-world use at this stage. In our Limitations section (Appendix A) and Impact Statement (Appendix B), we discuss the potential concern about bias (not unique to our approach but inherent in nearly all existing LLMs) and affirm that we did not comprehensively analyze potential demographic biases inherent in the data. Our evaluation dataset is constructed by randomly sampling a comprehensive set of Reddit posts and ShareGPT conversations, which ensures that our method is representative of the data distribution but does not introduce or remove any particular biases. We will add further discussions in the paper to more explicitly state that any future deployment of this method should include thorough analysis and mitigation of potential demographic bias, especially when applied to data from vulnerable groups.
>
> >Dataset Size and Scope: 220 documents might be considered modest for training robust LLM-based reasoning systems. The maximum of 10 disclosures per post is also a limitation for more extensive personal profiles. LLaMA-generated data may carry stylistic or topical biases.
>
> We thank the reviewer for their concerns. Due to the cost of annotation, we intentionally focused on atomic user posts. This required significant manual effort to ensure reliable gold-standard privacy risk estimates. Notably, the distribution of disclosures in our dataset is usually between 3–6 disclosures per post, closely matching the natural distribution of personal information across various subreddits, including from longer Reddit posts. We do not anticipate any fundamental limitations preventing BRANCH from generalizing to texts with more than 10 disclosures, though we have not extensively tested this case, as it is rare in naturally occurring data. We will clarify the scope of our dataset in the paper.
>
> We include a portion of synthetic data in our paper, solely for the purpose of responsible research, such that we can publicly share this annotated data. We use 50 synthetic documents solely for training; all evaluations are conducted exclusively on human-written posts.  LLaMA-generated data may carry distributional biases, so annotators extensively refined the synthetic posts to better match the content and tone of real disclosures, using human-written posts as references. When finetuned on the training set data without synthetic posts, we observe a drop in range accuracy to below 40%. In contrast, models of the same size finetuned with the synthetic data show substantial performance gains over both prompt-based inference and training on real data alone, suggesting that this augmentation strategy supports generalization rather than hinders it.
>
> >The methodology relies heavily on manual annotation of disclosures, Bayesian structures, and equations, which limits scalability and reproducibility. While inter-annotator agreement is reported (ρ = 0.916), constructing Bayesian networks with conditional dependencies is subjective and nontrivial.
>
> We agree with the reviewer on raising this point about scalability. Due to the need to construct a high-quality benchmark dataset, manual annotation is necessary to ensure reliable ground-truth labels for evaluation. Importantly, however, our proposed LLM-based method does *not* rely on manual annotations for inference.
> While constructing Bayesian networks involves some subjectivity, multiple valid network configurations yield similar estimations of the final k-values, as annotators’ predicted k-values were generally within an order of magnitude of each other. This is why we observe high inter-annotator agreement (ρ = 0.916), despite the flexibility in network construction. This consistency suggests that our method is robust to variations in network construction.
> >How was the “gold” k value validated in cases where no ground truth exists? For many Reddit or ShareGPT examples, ground truth is unclear. What confidence do we have in those values?
>
> In our current setup, we are assessing how closely LLM-based methods approximate human experts. Human gold standard labels are validated using double annotation and with the validation experiment in Section 4.1 (lines 184-194), which shows that for interest groups similar to those found in Reddit and ShareGPT, the human annotation methodology exhibits very low error when compared to the ground truths. While a perfect ground truth is still unattainable for some cases, we have made every effort to construct the most accurate evaluation data feasible. Authors of the paper also manually review the annotation process to ensure the quality of the gold k value. We will add to the paper to acknowledge this inherent limitation of working with real-world data in this challenging setting.
> >Why not use ground-truth census APIs or structured data sources to improve query answers? Could retrieval-augmented models improve robustness?
>
> This is a great point! We test RAG models in Appendix M (lines 999 onwards) and find that RAG systems only marginally improve performance, indicating that LLM internal knowledge is effective. We acknowledge that temporal drift is a concern to consider for possible deployment, but this temporal drift of demographic data would occur over a period of several years.
> >Are there any error modes where BRANCH performs worse than CoT?
>
> We did not identify specific error modes where BRANCH consistently performs worse than CoT. We do an error analysis in the paper (included in Appendix K) and find that the errors caused by BRANCH are similar to error types commonly found in CoT.

---

> > ### Comment · Reviewer_9q4K · 2025-08-04
> >
> > Thanks for taking the time to answer my questions. I am happy with the response and look forward to the updates in the paper. I have increased my score

---

### Official Review · Reviewer_TWWc · 2025-07-03

**Clarity:** 4
**Significance:** 3
**Originality:** 3
**Rating:** 4
**Confidence:** 4

**Summary:**

The paper proposes a novel method, BRANCH, to use LLMs for probabilistic reasoning to estimate the privacy risk of user-provided documents. The authors create a dataset using Reddit posts and ShareGPT conversations, along with a synthetic set generated using LLaMA-3.1-Instruct-8B and T5-large models to make the data publicly shareable. The proposed method decomposes the joint probability of personal attributes using a Bayesian network, with disclosure categories labeled automatically by a model or manually by human annotators. They evaluate the method using Spearman's rank correlation between the predicted and gold-standard k-values, and with a log error metric over the k-values, motivated by the wide variance in population sizes.

**Questions:**

1. Do you observe any systematic bias across demographic categories? How do you ensure that LLM-internal demographic priors do not skew both training and evaluation?
2.  Which stages (e.g., dependency modeling, subquery breakdown) are most critical to performance? Could a simplified variant retain similar accuracy? BRANCH includes many modular components, so I was curious to see if we can simplify them for cheaper cost to implement the method.
3. The paper reports a high Spearman correlation (p=0.916) between annotators, which suggests stable relative ranking. Could you provide additional insight into the actual variance or range of K values across different disclosure subsets for the same post? e.g., how much does k fluctuate depending on disclosure inclusion?

**Ethical Concerns:**

["NO or VERY MINOR ethics concerns only"]

**Final Justification:**

Responses the authors addressed are acknowledged. They are helpful to understand paper structures and how the responses are located within the paper, but it does not mean that the major weaknesses are addressed or mitigated. Hence, I keep my original positive score.

**Limitations:**

W5 is not a weakness the authors are required to address within this paper, but rather limitations that are inherent to this class of approaches. Unless the authors choose to adopt a theoretically grounded framework, these limitations may not be easily resolved.

**Quality:**

3

**Strengths And Weaknesses:**

## **Strengths**
1. Potential to be interpretable with bayesian factorization; conditioning on plausible priors (e.g., percentage of women in tech) sounds valid intuition than less tractable queries. Those query decomposition would contribute to capture more localized awareness of uncertainty for low-confidence answers.
2. Original problem framing; the work provides a unique data curator view of privacy into a data contributor view " how unique am I given what I said online?". This enables bayesian approaches with conditional probabilities of each subquery factors.
3. Practical settings; the authors create the synthetic dataset with Reddit to address the privacy issues in realistic settings. IRB approval obtained and the sensitive data is handled via T5 model generation and PII redaction.
4. Strong experimental results
## **Weaknesses**
1. Engineering heavy; the proposed method relies on a large number of heuristic components from disclosure selection to subquery decomposition to answer estimation. While effective, the final pipeline seems to be more of a procedural engineering than a coherent algorithmic framewrk.
2. Heavy reliance on LLMs;
- Both the estimations of conditional probabilities and the creation of benchmark dataset to evaluate the method are based on LLM responses. This makes the method critically exposed to LLM bias, even though the authors attempt to mitigate this by involving human annotations.
- LLM guesses are treated as statistical inference. The estimation of conditional probabilities via LLM generations lacks rigorous grounding. Such an approach falls into an assumption that the the LLM internal demographic believes as a valid values to model probabilistic estimates for k-values.
3. Some of those human involvement is also heuristic. Esp. the selection of disclosure appears heuristic, and despite their trying to estimate the variance of the disclosures, it still remains vulnerable to cherry-picked selections.
4. The prompt-based estimation is not quite innovative -- e.g., asking LLMs to output statistical answer or LLM-as-query-answerer is pretty common in current research trends.
5. No theoretical contribution; this limits its potential to generate deeper insights for future works from a core ML standpoint.

---

> ### Author Rebuttal · Authors · 2025-07-31
>
> >Both the estimations of conditional probabilities and the creation of the benchmark dataset rely on LLM responses. This makes the method critically exposed to LLM bias, even though the authors attempt to mitigate this through human annotations. LLM guesses are treated as statistical inference. The estimation of conditional probabilities via LLM generations lacks rigorous grounding and risks assuming that the LLM's internal demographic beliefs are valid for modeling probabilistic estimates.
>
> We thank the reviewers for raising this important concern. We would like to clarify that the benchmark dataset used to evaluate our method is constructed through careful human annotation, not generated by LLMs. As stated in lines 160–163, human annotators created the reference Bayesian networks and derived ground truth probabilities, drawing from real-world sources such as census data and public university statistics. We will revise the introduction to make this distinction more explicit.
>
> >Do you observe any systematic bias across demographic categories? How do you ensure that LLM-internal demographic priors do not skew both training and evaluation?
>
> This is a very interesting point. We discussed this in the Limitations section (Appendix A) and Impact Statement (Appendix B) in our paper. We do not observe obvious biases against specific demographic groups in our dataset; however, there could be some potential bias and potential LLM-internal demographic priors that come with the base LLMs we used in our method. This challenge is not unique to our approach, but shared by many if not all LLM-based inference methods, including chain-of-thought prompting baselines discussed in our paper. In our study, we sample from Reddit and ShareGPT following established data curation practices and treat LLM responses as approximate statistical surrogates. For evaluation, we compare the LLM-generated estimations to human-annotated ground truth in Section 5.2, ensuring that LLM-internal biases do not affect the model evaluation. A comprehensive audit of all possible sources of bias (e.g., gender, race, geography, education level, age, culture) would require significantly more data and human annotation resources than what is feasible within the scope and budget of this academic study. We acknowledge this as an important direction for future work. Our research is exploratory and academic in nature, aimed at advancing privacy-preserving methods for protecting user data. It is not intended for direct real-world deployment without further safeguards and a deeper understanding and mitigation of potential biases.
>
> >Some of those human involvement is also heuristic. Esp. the selection of disclosure appears heuristic, and despite their trying to estimate the variance of the disclosures, it still remains vulnerable to cherry-picked selections.
>
> We thank the reviewer for their comments and concerns. We would like to clarify that our human annotation process, including the selection of disclosures, is guided by prior work [1], which defines personal disclosures and provides a structured label schema, as mentioned in line 160 and Appendix C. We carefully adhere to this schema and consider a general threat model that accounts for all utterances of personal information within a post, so annotators do not filter out a subset of personal information in the text based on a heuristic process. Additionally, we doubly annotate a portion of our data to assess human consistency and annotator variance. We apologize for the confusion and will provide the full annotation guidelines in the Appendix.
>
> The human involvement for selecting disclosures is only for filtering redundant information (e.g., multiple mentions of the same occupation in a post) that do not add statistical value to the privacy risk estimate. Regarding the variance during human annotation, this largely arises from the process of Bayesian network structure elicitation, not from the disclosure content itself. This includes variation in edge directionality and topological ordering of the variables, which may result in different conditional probability tables used to estimate k. To further assess model robustness, future work should look at user studies and explicitly examine performance on edge cases, including adversarial scenarios.
>
> [1] Dou, Y., Krsek, I., Naous, T., Kabra, A., Das, S., Ritter, A., & Xu, W. (2023). Reducing Privacy Risks in Online Self-Disclosures with Language Models. ArXiv, abs/2311.09538.
>
> >The prompt-based estimation is not quite innovative -- e.g., asking LLMs to output statistical answer or LLM-as-query-answerer is pretty common in current research trends.
>
> We thank the reviewer for this observation and acknowledge the existing research that also utilizes LLMs as direct query-answering tools. We would like to clarify that our contribution goes beyond statistical query estimation with LLMs, as we introduce a probabilistic framework that decomposes joint probabilities into a structured Bayesian network and estimates the occurrence rates of personal information using conditional probability distributions. Large Language Models perform structured probabilistic reasoning, which is less explored in the literature, especially in practical domains with grounded human-annotated evaluations like user-level privacy risk estimation. We will clarify this distinction more explicitly in the paper.
>
> >No theoretical contribution; this limits its potential to generate deeper insights for future works from a core ML standpoint.
>
> We acknowledge this point mentioned by the reviewer and agree that our paper does not contain contributions to ML theory. Instead, our work is primarily empirical, similar to a lot of recent work on LLMs and deep learning. Our task is application-driven, aiming to address a critical gap in user-side privacy support by using the probabilistic reasoning capabilities of LLMs, which we believe opens up useful directions for applied research in other tasks.
>
> >Which stages (e.g., dependency modeling, subquery breakdown) are most critical to performance? Could a simplified variant retain similar accuracy? BRANCH includes many modular components, so I was curious to see if we can simplify them for a cheaper cost to implement the method.
>
> We thank the reviewer for this question! We run ablation studies (included in Appendix J) and find that the dependency modeling is the most critical component for performance, largely due to its key importance in constructing a Bayesian Network to estimate the privacy risk. The heuristic modules marginally improve performance, so a simplified variant, such as removing the components of subquery breakdown would result in a cheaper inference cost but comparable performance, especially for the simpler cases with less number of disclosures. In principle, with a substantially larger set of human-annotated data (albeit requiring significant upfront human annotation effort), it would be possible to fine-tune an end-to-end model that eliminates some intermediate components and significantly reduces inference-time computational cost.
>
> >The paper reports a high Spearman correlation (p=0.916) between annotators, which suggests stable relative ranking. Could you provide additional insight into the actual variance or range of K values across different disclosure subsets for the same post? e.g., how much does k fluctuate depending on disclosure inclusion?
>
> In our doubly annotated data, the variance mainly arises from the differing Bayesian networks that are constructed to solve for k, rather than from differing disclosure subsets. On average, we see an average log difference between the human gold k predictions of 1.56. This variance indicates that annotators generally agree within the same order of magnitude, supporting the observed high Spearman correlation and low variance.

---

> > ### Comment · Reviewer_TWWc · 2025-08-06
> >
> > Thank you for taking time to address the concerns above. The responses were helpful, and addressed most concerns. Some inherent limitations such as bias, annotation subjectivity, and reliance on LLM priors understandably remain unresolved, as they are challenging to eliminate entirely within the scope of this type of work. That said, the new clarifications provided during the rebuttal (e.g., synthetic data justification, personalized threat models, evaluation scope) are helpful and should be incorporated into the final version to improve transparency and clarity.
> >
> > I remained my positive scores.

---

### Note · Authors · 2025-08-14

We thank all reviewers for their thoughtful feedback and are encouraged by the initial scores (4, 4, 4, 5). After discussion, we appreciate that two reviewers who gave a 4 indicated they are considering increasing their scores

In our discussion, reviewers highlighted the novelty of the task and problem framing, shifting privacy risk assessment from dataset owners to end-users with an interpretable k-value. The creation of a high-quality, privacy-preserving evaluation dataset with IRB approval was seen as valuable, alongside comprehensive experiments and error analyses. Finally, reviewers highlighted that the LLM methodology, which used Bayesian networks and conditioned on plausible priors, was regarded as well-structured and interpretable, with meaningful contributions to practical privacy protection and potential real-world impact.

We will address the reviewers’ comments and concerns by updating our paper to include the full annotation guidelines of our dataset for reproducibility and further experimental details on model selection and hyperparameter choices. We have also conducted the new experiments requested by the reviewers, examining LLM consistency in our privacy task across multiple runs and providing a computational analysis table illustrating how our method scales with input complexity. We will incorporate these results in our paper.

---

### Decision · Program_Chairs · 2025-09-17

**Decision:**

Accept (poster)

**Comment:**

The paper introduces BRANCH, a new method that uses LLMs to estimate the privacy risk of a text. The main idea is to estimate a k-value, which represents the number of people in the world who share the personal attributes mentioned in a document.
The BRANCH method breaks down this complex problem into smaller, manageable steps. It treats the personal information as random variables in a Bayesian network to account for dependencies between different attributes (e.g., age, location, health conditions). It then uses an LLM to individually estimate the probability of each attribute. Finally, it combines these individual probabilities based on the structure of the Bayesian network to calculate the final k-value.

This approach is shown to be more accurate than traditional methods, particularly for documents with multiple personal attributes. The paper also highlights the importance of quantifying LLM uncertainty, as predictions with high variance were found to be less accurate.

The reviewers recommend accepting the paper given its originality and novelty, strong methodology, high-quality dataset and evaluation, and practical significance.

The reviewers also raised several weaknesses, including the computational cost, the reliance on LLMs, and limitations of the dataset and threat model. However, the authors' rebuttals were considered comprehensive and satisfactory, addressing most of these concerns.

In light of the above, I support accepting the paper.